# Genomic characterization of the HER2-enriched intrinsic molecular subtype in primary ER-positive HER2-negative breast cancer

Lennart Hohmann [1,2], Kristin Sigurjonsdottir[1,2], Ana Bosch Campos[1], Deborah F. Nacer [1,2], Srinivas Veerla [1,2], Frida Rosengren[1], Poojaswini Thimmaraya Reddy [2], Jari Häkkinen [1], Nicklas Nordborg[1], Johan Vallon-Christersson [1], Yasin Memari[3], Daniella Black[3], Ramsay Bowden [3], Helen R. Davies [3], Åke Borg [1], Serena Nik-Zainal [3] & Johan Staaf [1,2] ✉

ER-positive/HER2-negative (ERpHER2n) breast cancer classified as PAM50 HER2-enriched (ERpHER2n-HER2E) represents a small high-risk patient subgroup. In this study, we investigate genomic, transcriptomic, and clinical features of ERpHER2n-HER2E breast tumors using two primary ERpHER2n cohorts comprising a total of 5640 patients. We show that ERpHER2n-HER2E tumors exhibit aggressive clinical features and poorer clinical outcomes compared to Luminal A and Luminal B tumors. Furthermore, ERpHER2n-HER2E breast cancer does not consist of misclassified or HER2-low cases, has little impact of *ERBB2*, is highly proliferative and less ER dependent than other luminal subtypes. It is not an obvious biological entity but is nevertheless associated with potentially targetable molecular features, notably a high immune response and high *FGFR4* expression. Strikingly, molecular features that define the HER2E subtype in luminal disease are also consistent in HER2-positive disease, including an epigenetic mechanism for high *FGFR4* expression in breast cancer.

Breast cancer is the most common malignancy amongst women worldwide[1], and a highly heterogeneous disease encompassing numerous tumor types that are characterized by specific molecular profiles and morphologies, bearing distinct clinical implications. Clinically the disease is divided into subgroups based on the assessment of three main treatment predictive biomarkers: protein overexpression of the estrogen receptor alpha (ER) and progesterone (PR) receptors, and overexpression/gene amplification of the human epidermal growth factor receptor 2 (HER2, a tyrosine kinase receptor

encoded by the gene *ERBB2*). The subgroups defined by these markers are triple-negative breast cancer (TNBC, negative for all three markers), HER2-amplified disease further stratified by ER-status (ER-positive (ERpHER2p) and ER-negative (ERnHER2p)), and ER-positive HER2-negative tumors (ERpHER2n). ERpHER2n tumors constitute the largest subgroup and are also collectively referred to as luminal due to expressing typical proteins of luminal epithelial cells[2].

Gene expression profiling of breast cancer has identified intrinsic molecular subtypes[3] that improve the representation of breast

[1]Division of Oncology, Department of Clinical Sciences Lund, Lund University, Lund, Sweden. [2]Division of Translational Cancer Research, Department of Laboratory Medicine, Lund University, Lund, Sweden. [3]Academic Department of Medical Genetics, School of Clinical Medicine & Early Cancer Institute, University of Cambridge, Cambridge, UK. ✉e-mail: johan.staaf@med.lu.se

cancer's inherent clinical and biological heterogeneity. The currently most used multi-gene subtyping signature is PAM50, which stratifies tumors into five subtypes based on the expression of a 50-gene subset and nearest centroid correlation: Basal-like (Basal), HER2-enriched (HER2E), Luminal A (LumA), Luminal B (LumB), and Normal-like (Normal)[4]. When applied to population-representative breast cancer cohorts the dominant PAM50 subtypes in ERpHER2n tumors are LumA (47%) and LumB (30-40%), however small subgroups of both Basal and HER2E tumors are also observed[5,6].

Despite being clinically determined as HER2-negative, 7-9% of ERpHER2n tumors are PAM50 subtyped as HER2E[5,7,8]. Studies have indicated that the HER2E subtype is associated with poor outcome in early-stage disease[9–12], shorter progression-free survival with endocrine therapy[13,14], chemo-sensitivity, resistance to CDK4/6 inhibition[10], as well as anti-HER2 sensitivity within metastatic disease[15]. In the advanced setting, the HER2E subtype has been reported to be more prevalent compared to the primary setting[16], potentially due to biological changes associated with the intrinsic evolution of the tumor or as a result of selective pressure from earlier treatment[17].

In early-stage disease, ERpHER2n-HER2E tumors are observed both in patients treated with only adjuvant endocrine therapy (ET) and in patients treated with combined adjuvant chemotherapy and endocrine therapy (CT + ET)[5]. With respect to anti-HER2 therapy, it has been suggested that the HER2E subtype may be an overall better predictor of HER2-targeted therapy response than the regularly employed clinical assessments (i.e., immunohistochemistry (IHC) and fluorescence in situ hybridization (FISH))[18,19] and that EGFR/HER2 tyrosine kinase inhibition might be of value for ERpHER2n-HER2E tumors[10], although this remains to be proven in clinical trials. A subclassification that has recently gained considerable attention in breast cancer is the so called HER2-low group (HER2 IHC 2 + /ISH- or IHC 1 + ), based on the introduction of antibody-drug conjugates (ADCs) targeting HER2-expressing cells (e.g., the FDA-approved ADC Trastuzumab Deruxtecan). With respect to PAM50 subtypes, the HER2-low group is however a mix of subtypes and is not equivalent to the ERpHER2n-HER2E patient subgroup[20]. Despite its apparent association with poor patient outcome in both primary and advanced disease, the HER2E subtype in ERpHER2n disease currently holds no implications for clinical treatment decisions, besides a marker of aggressive disease, likely due to a lack of insight into its characterizing features.

In this work we comprehensively characterize ERpHER2n-HER2E breast cancer and compare it to other ERpHER2n patient groups, as well as to ERpHER2p disease, based on clinicopathological and molecular features on RNA and DNA level. We evaluate the response of patients to conventionally administered therapies and identify features that can potentially be used to inform future treatment decisions.

## Results

The current study aimed to perform an in-depth clinicopathological and molecular characterization of ERpHER2n-HER2E tumors (hereafter referred to as HER2E) based on comparisons with ERpHER2n PAM50-classified LumA and LumB tumors and with ERpHER2p tumors. Figure 1 shows an outline of the performed analysis and used cohorts.

### HER2E disease and clinicopathological variables

The clinicopathological characterization of HER2E tumors in the SCAN-B and METABRIC cohorts is provided in Table 1. Comparisons demonstrated that HER2E tumors were significantly larger in size and associated with higher grades than LumA tumors in both cohorts. SCAN-B data indicated that HER2E patients tended to be more often PR-negative and younger than LumB patients, although this pattern was not observed in METABRIC cases (PR data not available). The frequency of HER2-low samples was not significantly different between HER2E, LumA, and LumB for SCAN-B cases, while in METABRIC the HER2E group was associated with a lower frequency of HER2-low

classified cases compared to both LumA and LumB cases. 21-gene Recurrence Score (RS) classification based on RNA-sequencing FPKM data labelled 96% of SCAN-B HER2E tumors as high-risk cases, confirming it to be a high-risk subgroup of breast cancer patients compared to both LumA and LumB patients in pair-wise comparisons (Supplementary Table S1). Risk of Recurrence (ROR) score classification based on RNA-sequencing derived estimates from Staaf et al.[6] also confirmed HER2E tumors to be high-risk cases compared to both LumA and LumB patients in pair-wise comparisons (Supplementary Table S1). Of 84 HER2E tumors having both classifications, 89.3% were classified as high-risk by both gene signatures. Comparisons with available IC10 gene expression classification[21] in METABRIC demonstrated that HER2E cases were not associated with any specific IC10 class. Clinicopathological characteristics of reviewed SCAN-B cases are available in Supplementary Table S2.

### HER2E disease and patient outcome

Survival analyses were performed in both SCAN-B and METABRIC stratified by adjuvant therapy (ET or CT + ET, Fig. 2) and were based on invasive disease-free survival (IDFS) and recurrence-free interval (RFI), respectively, as well as overall survival (OS) (both cohorts). In the ET group, Kaplan-Meier analysis and univariate Cox regression demonstrated that patients with HER2E tumors had worse outcomes compared to patients with LumA and LumB tumors (Hazard ratios (HR) with LumA as reference: SCAN-B (LumB=1.7, HER2E = 4.0); METABRIC (LumB=1.6, HER2E = 2.1); Fig. 2A, B). Similar patterns were observed in the CT + ET treatment group, where HER2E patients were also associated with significantly worse outcomes (HRs with LumA as reference: SCAN-B (LumB=1.2, HER2E = 2.8); METABRIC (LumB=2.0, HER2E = 5.6); Fig. 2E, F).

The multivariate Cox regression analyses in the SCAN-B ET and CT + ET treatment groups, as well as in the METABRIC CT + ET treatment group, further demonstrated that the HER2E subtype provided independent prognostic information for IDFS and RFI outcomes in models with patient age, lymph node status, tumor size and tumor grade as covariates (Fig. 2C, G, H). Unlike in SCAN-B, this observation was not significant in METABRIC ET-treated patients, hampered by the limited subgroup size (Fig. 2D).

Survival analysis with OS as clinical endpoint demonstrated a shorter survival of HER2E patients in the ET treatment group within both SCAN-B and METABRIC (HRs with LumA as reference: SCAN-B LumB=1.9 and HER2E = 3.1, METABRIC LumB=1.4 and HER2E = 1.6; Supplementary Fig. S1A, B). The independent prognostic value of the HER2E subtype remained also in multivariate Cox Regression analyses in both cohorts (Supplementary Fig. S1C, D). HER2E patients treated with CT + ET did not have significantly different overall survival intervals than other luminal subtypes, potentially due to small patient numbers (Supplementary Fig. S1E–H).

### HER2E tumors are proliferative and immune-inflamed

Analysis of expression patterns for six biological metagenes[22] in the SCAN-B cohort showed significant differences between the HER2E and LumA subtypes for the basal, immune response, steroid response, lipid, and mitotic progression metagenes (Fig. 3A-C; Supplementary Fig. S2A–C). Significant metagene score differences were also observed between the HER2E and LumB subtypes for the basal, stroma, steroid response, and immune response metagenes. Notably, the HER2E group was distinctly associated with a lower steroid response and higher immune response compared to other PAM50 subtypes, which was corroborated by findings in the METABRIC cohort (Supplementary Fig. S3B, C).

We used CIBERSORTx to delineate in silico different cell type proportions, acknowledging its limitations in accuracy. The immune response metagene scores correlated strongest with CIBERSORTx CD8 T-cells and Monocyte estimates based on 4377 analyzed cases

**A**

**B**

**Fig. 1 | Study overview detailing cohorts used for analyses within ERpHER2n and ERpHER2p disease.** . **A** Analyses comparing ERpHER2n-HER2E to the PAM50 subtypes LumA and LumB within ERpHER2n disease. **B** Analyses comparing ERpHER2n-HER2E to ERpHER2p disease subtyped as PAM50 HER2E or another subtype. ACT adjuvant chemotherapy, TME tumor microenvironment, DEG Differentially expressed gene, CNA Copy number alteration.

**Table 1 | Clinicopathological characteristics of the intrinsic luminal breast cancer subtypes**

| Variable | HER2E (ref) | SCAN-B LumA | LumB | HER2E (ref) | METABRIC LumA | LumB |
|---|---|---|---|---|---|---|
| N | 89 (2%) | 3049 (68%) | 1349 (30%) | 58 (6%) | 601 (60%) | 340 (34%) |
| Patient age (years) | 64 ± 14.4 | 65.3 ± 12.2 | 67.5 ± 12.8* | 63.8 ± 11.9 | 63.2 ± 12.4 | 65.7 ± 11.3 |
| Tumor size (mm) | 21.1 ± 9 | 18.4 ± 12.1** | 21.6 ± 12.3 | 29.1 ± 18 | 23.9 ± 12.1* | 27.5 ± 14.5 |
| positive PR status: | 57 (64%) | 2701 (89%)**** | 1076 (80%)** | – | – | – |
| Tumor grade | | | | | | |
| NHG1 | 0 (0%) | 834 (28%)**** | 73 (6%)* | 2 (4%) | 108 (19%)**** | 15 (5%) |
| NHG2 | 35 (43%) | 1929 (64%)**** | 593 (45%)* | 20 (38%) | 329 (57%)**** | 143 (43%) |
| NHG3 | 47 (57%) | 245 (8%)**** | 640 (49%)* | 31 (58%) | 137 (24%)**** | 171 (52%) |
| LN status | | | | | | |
| N0 | 49 (58%) | 1994 (67%) | 802 (60%) | 35 (60%) | 334 (56%) | 169 (50%) |
| N+ | 35 (42%) | 1001 (33%) | 526 (40%) | 23 (40%) | 267 (44%) | 171 (50%) |
| N reviewed | 27 (30%) | 508 (17%) | 369 (27%) | – | – | – |
| HER2-low[A] | 22 (81%) | 436 (86%) | 316 (86%) | 13 (22%) | 257 (43%)** | 137 (40%)* |
| IC10 class | | | | | | |
| IC-1 | – | – | – | 7 (12%) | 8 (1%) | 53 (16%) |
| IC-2 | – | – | – | 5 (9%) | 21 (3%) | 26 (8%) |
| IC-3 | – | – | – | 7 (12%) | 180 (30%) | 36 (11%) |
| IC-4ER+ | – | – | – | 7 (12%) | 88 (15%) | 26 (8%) |
| IC-5 | – | – | – | 0 (0%) | 0 (0%) | 0 (0%) |
| IC-6 | – | – | – | 8 (14%) | 15 (2%) | 30 (9%) |
| IC-7 | – | – | – | 8 (14%) | 99 (16%) | 34 (10%) |
| IC-8 | – | – | – | 7 (12%) | 168 (28%) | 79 (23%) |
| IC-9 | – | – | – | 8 (14%) | 21 (3%) | 43 (13%) |
| IC–10 | – | – | – | 1 (2%) | 1 (0%) | 13 (4%) |

A) For SCAN-B cases HER2-low classification was only possible in reviewed cases.
Percentage definitions: Across subtypes (N); within subtype (PR status, Tumor grade, LN status, N reviewed, HER2-low, IC10 class). Pairwise comparisons of HER2E vs. LumA and LumB, respectively.
Statistics: Fisher's exact tests for variables PR status, Tumor grade, LN (lymph node) status, and HER2-low frequency; two-sided t-tests for variables Patient age and Tumor size. Significance annotation: * ≤0.05; ** ≤0.01; *** ≤0.001; **** ≤0.0001. All reported p-values are two-sided.

(Fig. 3D). Consistently, HER2E tumors had statistically significant higher estimated proportions for these cell types compared to LumA and LumB tumors, as well as for NK cells, but not B-cells (Fig. 3E). Additionally, we observed a higher expression of the PD-L1 encoding gene *CD274* within HER2E (Fig. 3F). To complement CIBERSORTx and investigate the specific cellular composition within HER2E tumors, we analyzed in situ estimates of tumor infiltrating lymphocyte counts derived from single-cell METABRIC data, as presented by Danenberg et al.[23]. After combining immune cell phenotype counts based on Spearman correlation (Supplementary Fig. S4A) our results indicate that the tumor microenvironment of HER2E tumors tends to be more enriched in immune cells (combined single cell phenotype counts of CD4 + T-cells, CD8 + T-cells, Macrophages, and B-cells) compared to LumA and LumB tumors. The differences in immune cell proportions were more evident when comparing PAM50 subtypes regardless of clinical group (Supplementary Fig. S4B), as the number of HER2E subtyped ERpHER2n cases was low in the METABRIC subset analyzed by Danenberg et al. (Supplementary Fig. S4C).

**Expression of *ESR1* and *ERBB2* in HER2E disease**
*ERBB2* mRNA expression was investigated to address the questions of potential sample misclassification, i.e., whether HER2E cases could be misclassified *ERBB2*-amplified cases, and whether HER2E samples as a group were characterized by an elevated *ERBB2* mRNA expression (e.g., by some other mechanism than gene amplification). Sample misclassification assessment was only performed in the SCAN-B cohort due to falsely classified HER2-negative samples (i.e., *ERBB2*-amplified) being by definition impossible in the METABRIC cohort, where HER2-status was determined using copy number array data. While we

observed a statistically significant difference in the expression of *ERBB2* between HER2E and LumB tumors, its magnitude suggests it being biologically inconsequential (Fig. 3G, Supplementary Fig. S3E). Notably, of all 89 SCAN-B HER2E cases only six (6.7%) were suspected as potentially misclassified from being extreme outliers (Fig. 3G). None of these six samples had associated WGS data, so *ERBB2* amplification status could not be definitively resolved. The steroid response meta-gene findings and the lower rate of PR-positivity in HER2E tumors were confirmed by assessing the expression of *ESR1* (which is not included in the steroid response metagene) showing a significantly lower expression of *ESR1* in HER2E tumors in both cohorts (Fig. 3H; Supplementary Fig. S3D).

**Transcriptional patterns of HER2E disease**
Of 19644 genes in the SCAN-B cohort, 5246 (26.7%) were identified to be differentially expressed between HER2E and LumA cases, and 3465 (17.6%) between HER2E and LumB cases. In METABRIC, 24368 genes were assessed, of which 1923 (7.9%) were differentially expressed between HER2E and LumA cases, and 1525 (6.3%) between HER2E and LumB cases. By determining which genes were differentially expressed in both cohorts, core sets of differentially expressed genes (cDEGs) were defined. In the comparison between HER2E and LumA, differentially expressed genes constitute the LumA cDEG set (*n* = 1100), while in the HER2E-LumB comparison, differentially expressed genes form the LumB cDEG set (*n* = 522). The HER2E cDEG set includes genes significant in both pairwise comparisons (HER2E cDEG *n* = 258, Supplementary Dataset 1). Pathway enrichment analyses did not reveal substantial overlaps for the HER2E cDEG and LumB cDEG sets, while for the LumA cDEGs an enrichment of cell cycle-associated genes was

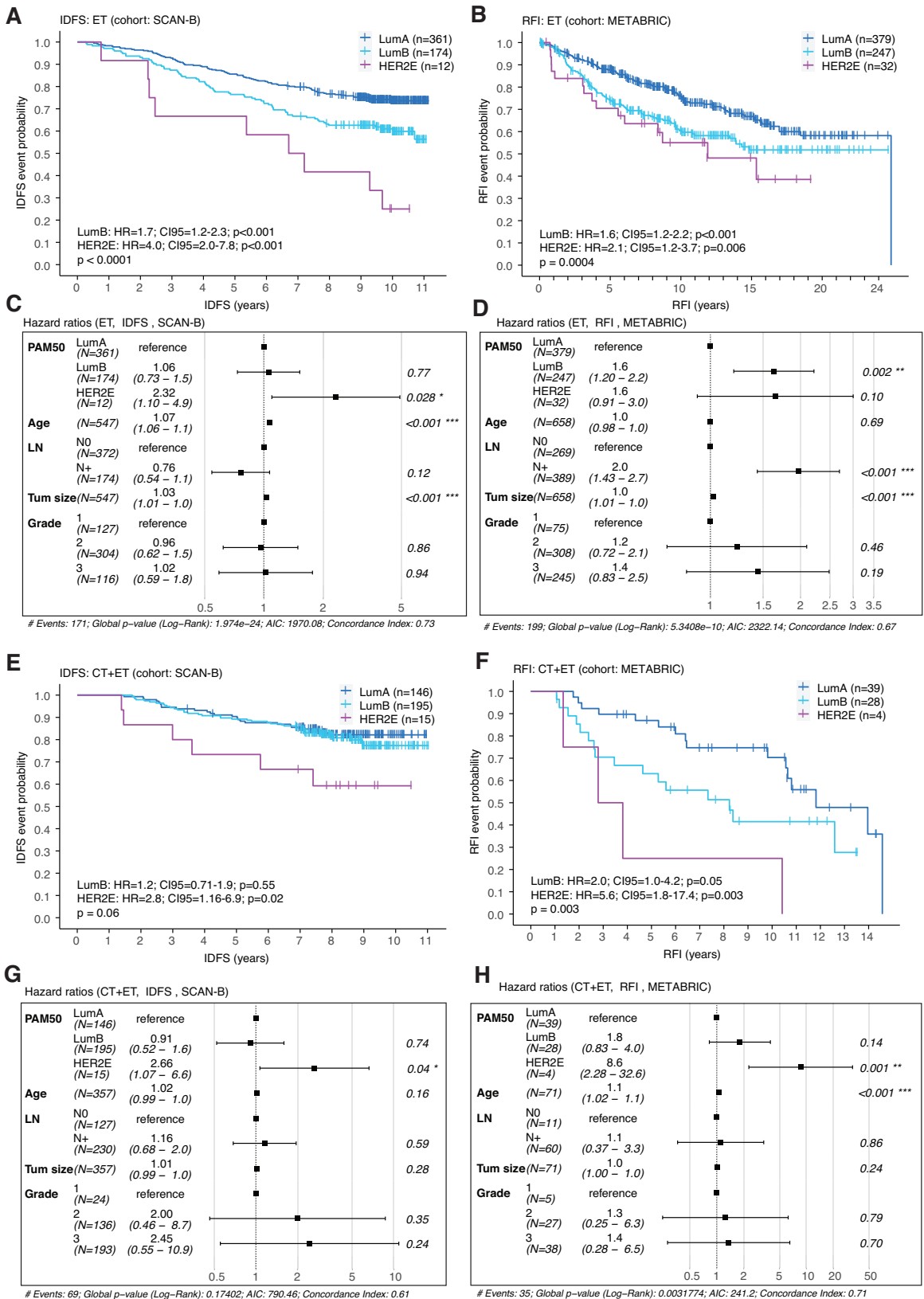

observed (Supplementary Fig. S2E). PAM50 genes found in the HER2E cDEG (*n* = 258) set included *CCNE1, BCL2, ESR1, SLC39A6, NAT1*, and *FGFR4*. The top HER2E cDEGs (absolute scaled mRNA expression difference of ≥1.5 to both LumA and LumB) were *ESR1, TBC1D9, CCDC170, RERG, IGF1R, FGFR4, BCL2, SOX11*, and *THSD4*.

To analyze the more general transcriptional distinctiveness of the HER2E group we performed unsupervised UMAP clustering of samples based on the scaled mRNA expression of all genes. This analysis demonstrated no concise clustering of HER2E samples, indicating that HER2E tumors are not distinctly different from LumA and LumB

**Fig. 2 | Survival analyses of PAM50 subtypes in ERpHER2n tumors in SCAN-B and METABRIC with IDFS and RFI as clinical endpoints. A** Kaplan-Meier analysis of SCAN-B luminal breast cancers treated with endocrine therapy (ET) including PAM50-based univariate Cox regression hazard ratios (HRs) and 95% confidence intervals (CI). **B** Kaplan-Meier analysis of METABRIC luminal breast cancers treated with ET including PAM50-based univariate Cox regression HRs and 95% CIs. **C** Multivariate Cox regression HRs with 95% CIs for SCAN-B cases treated with ET. **D** Multivariate Cox regression HRs with 95% CIs for METABRIC cases treated with ET. **E** Kaplan-Meier analysis of luminal SCAN-B breast cancers treated with the combination of chemotherapy and endocrine therapy (CT + ET) including PAM50-based univariate Cox regression HRs. **F** Kaplan-Meier analysis of luminal METABRIC breast cancers treated with CT + ET including PAM50-based univariate Cox regression HRs. **G** Multivariate Cox regression HRs with 95% CIs for SCAN-B cases treated with CT + ET. **H** Multivariate Cox regression HRs with 95% CIs for METABRIC cases treated with CT + ET. Differences in Kaplan-Meier survival curves were assessed by employing log-rank tests. In multivariate analyses lymph node status N0 and N+ indicate the absence or presence of regional lymph node metastases, respectively. Significance annotation: * ≤0.05; ** ≤0.01; *** ≤0.001; **** ≤0.0001. Source data are provided as a Source Data file.

tumors in terms of overall gene expression (Fig. 3J; Supplementary Fig. S3G). In addition, we plotted the HER2-enriched PAM50 centroid correlations against the LumA and LumB PAM50 centroid correlation for all HER2E samples to assess differences in the distinctiveness of the PAM50 HER2E subtype assignment. The results indicate that within HER2E cases, PAM50 HER2-enriched centroid correlations tend to be closer to LumB centroid correlations than LumA centroid correlations (Fig. 3K, L).

### Transcription binding factor analysis of HER2E core genes

To further analyze the 258 HER2E cDEG set we performed a correlation network analysis as originally described by Fredlund et al.[22]. Thereby two separate gene networks were identified, one upregulated in HER2E cases and the other one downregulated, although they were not substantially enriched for any pathway. Transcription factor binding analysis of the networks identified the binding site motif of the transcription factor GABPA to be present within all promotors of HER2E cDEGs. On the other hand, the copy number and mutational alteration landscape of the most affected transcription factors (including GABPA) did not show any consistent features within the 28 WGS analyzed HER2E cases (Supplementary Dataset 2).

### Mutational and driver gene characterization of HER2E disease

Differences in exposure to mutational signatures between ERpHER2n tumors classified as HER2E and LumA/LumB were analyzed in the WGS-profiled SCAN-B/BASIS patient sets. The HER2E subtype was associated with significantly lower exposure to the SBS1 and SBS5 mutational signatures than LumA and LumB tumors, but higher exposure to the APOBEC-related signatures SBS2 and SBS13 (Fig. 4A–D). APOBEC mutagenesis has been linked to a mutational phenomenon referred to as kataegis, which is a localized form of single-base substitution hypermutation[24]. Consistently, the frequency of tumors with at least one kataegis event was significantly higher in the HER2E subtype with 85.7%, compared to only 34.2% and 53.3% in LumA and LumB, respectively (Fisher's exact test vs. HER2E: LumA p = 4.048e-06; LumB p = 0.002).

We used the WGS-based HRDetect classifier to identify tumors with genome profiles indicative of homologous recombination deficiency (HRD)[25]. Notably, 17.9% of HER2E tumors were classified as HRDetect-high, compared to 6.8% in LumA and 9.5% in LumB, although the differences were not statistically significant (Fisher's exact test vs. HER2E: LumA *p* = 0.1; LumB *p* = 0.3). Comparisons of six structural rearrangement signature proportions identified a lower exposure to rearrangement signature 2 (predominantly characterized by dispersed translocations)[26] in the HER2E subtype compared to both the LumA and LumB subtype (Fig. 4E).

While *PIK3CA* mutation frequency did not differ significantly between HER2E (50%) and LumA (45.2%) or LumB (30.4%), the high frequency of *TP53* mutations was identified as a distinct feature of HER2E tumors (frequency of *TP53*-mutated cases: 60.7% in HER2E, 8.2% in LumA, and 17.1% in LumB; Fig. 4H). The frequency of *MYC* amplifications did not differ significantly between HER2E (25%) and LumB (17.1%) but was higher than in the LumA subtype (2.7%). *ERBB2* was significantly more frequently mutated in HER2E (14.3%) cases than in LumA (1.4%) and LumB (2.9%) cases, although the overall mutation frequency was still too low to be a group-defining trait (Fig. 4I). These findings were consistent with observations in the METABRIC cohort (Supplementary Fig. S5A, B and E).

### The HER2E copy number landscape

HER2E tumor genomes were found to be significantly more affected by copy number alterations compared to LumA tumors, but equivalent to LumB tumors (Fig. 4J, Supplementary Fig. S5C). Statistical comparison of copy number gain frequencies for 19982 genes across the genome identified 646 genes that differed between the HER2E and LumA subtypes, and 229 that differed between the HER2E and LumB subtypes (Supplementary Dataset 3). For copy number loss, 364 genes differed between the HER2E and LumA subtypes, but only two differed between the HER2E and LumB subtypes. These results demonstrate that on a global copy number level the HER2E and LumB subtypes do not differ substantially as illustrated by each group's genome-wide alteration profiles (Fig. 4L, Supplementary Fig. S5F). The mean absolute difference in alteration frequency of the statistically significant genes was equal to 29.3% and 24.5% for LumA and LumB, respectively (Fig. 4K).

### HER2E disease compared to HER2-amplified tumors

To compare ERpHER2n-HER2E tumors (HER2E) to ERpHER2p-HER2E and non-HER2E tumors (hereafter referred to as HER2p-HER2E and HER2p-NonHER2E, respectively) a set of analyses analogous to those done for the ERpHER2n subgroups (as outlined in Fig. 1B) was performed on the RNA (using the SCAN-B cohort) and DNA (using the METABRIC cohort) levels. In these analyses, the focus was to compare the HER2E-characterising features from the ERpHER2n analyses to ERpHER2p tumors. Firstly, none of the 258 HER2E cDEGs were differentially expressed between HER2E and HER2p-HER2E cases, while 95.7% (n = 247) were differentially expressed also between HER2E and HER2p-NonHER2E (Mann-Whitney U test with FDR correction p ≤ 0.05). Regarding expression of the immune response, mitotic progression, and steroid response metagenes there were no significant differences between HER2E and HER2p-HER2E cases, while there were significant differences compared to the HER2p-nonHER2E subgroup (Fig. 5A–C). The same observation was true for the comparison of *ESR1* and *FGFR4* mRNA expression, while *ERBB2* mRNA expression was as expected higher in both HER2p subgroups (Fig. 5D–F). No significant differences in the frequency of *ERBB2* and *TP53* mutations were observed (Fig. 5G, H). Furthermore, no concise clusters were observed, suggesting that none of these groups differ distinctively in terms of overall gene expression patterns (Fig. 5I). Akin to HER2E tumors, disregarding *ERBB2* amplifications, the mutational and amplification landscape of cancer drivers within HER2p-HER2E tumors was mainly characterized by high frequencies of predominantly missense mutations in *PIK3CA* (38.7%) and *TP53* (68.2%), as well as *MYC* amplifications (38.6%), thereby further corroborating the similarities between HER2E and HER2p-HER2E tumors (Fig. 5J).

### Epigenetic regulation of *FGFR4* in breast cancer

*FGFR4* was selected to be a prototypical high expressing gene for HER2E in the PAM50 subtyping scheme besides *ERBB2* and *GRB7*

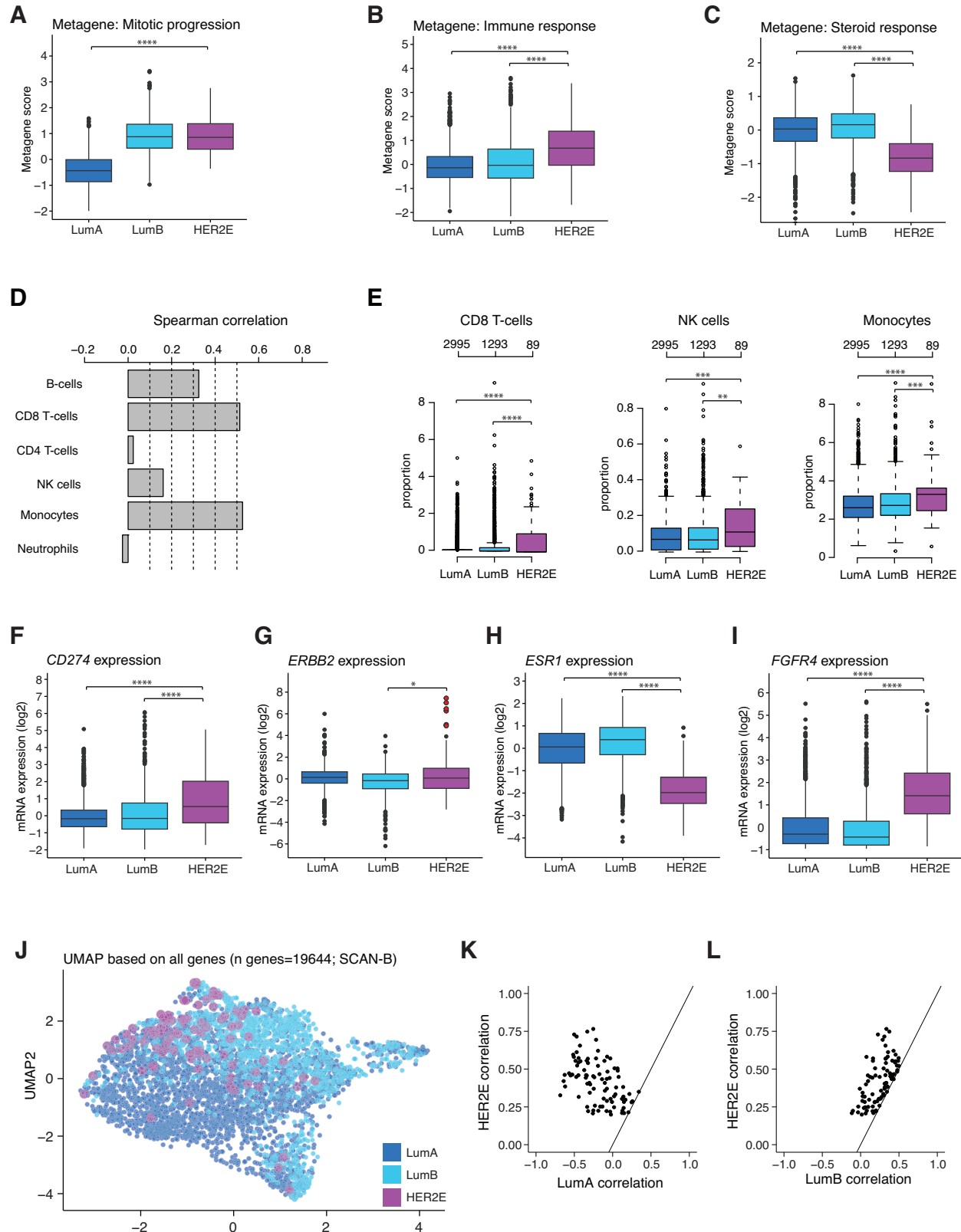

(both located in the minimal amplified region on 17q12)[4] and has been shown to have an equal or in ERpHER2n disease even slightly larger impact on PAM50 stability than the *ERBB2/GRB7* locus[27]. Yet, our analyses highlight three important aspects of *FGFR4* in breast cancer, i) that heterogeneity in expression within HER2E tumors exists (e.g., Fig. 3I), ii) that heterogeneity in expression exists

between PAM50 subtypes, with, e.g., high *FGFR4* expressing tumors also in the PAM50 LumA and LumB subtypes (Fig. 3I), and iii) that the gene does not appear driven by copy number alterations or somatic alterations as underlined by the total absence of called *FGFR4* amplifications or pathogenic somatic mutations in our WGS-analyzed ERpHER2n HER2E tumors.

**Fig. 3 | Transcriptomic analyses in the HER2-enriched subtype in SCAN-B.**
Analyses were based if not otherwise illustrated on 89 HER2E tumors, 3049 LumA tumors, and 1349 LumB tumors. **A** Mitotic progression metagene scores. **B** Immune response metagene scores. **C** Steroid response metagene scores. **D** Spearman correlation of immune metagene scores versus CIBERSORTx estimated cell type proportions for 4377 ERpHER2n SCAN-B tumors. **E** CIBERSORTx CD8 T-cells, NK cells, and Monocytes estimates versus PAM50 subtypes. B-cells estimates not depicted due to not exhibiting significant differences between subtypes. Some tumors lack data. **F** Scaled mRNA expression of *CD274*. **G** Scaled mRNA expression of *ERBB2* with potentially misclassified samples indicated in red (defined as extreme outliers, i.e., *ERBB2* mRNA expression > Q3 + IQR*3, where Q3 represents the third quartile and IQR the interquartile range). **H** Scaled mRNA expression of *ESR1*. **I** Scaled mRNA expression of *FGFR4*. **J** UMAP analysis based on FPKM expression data of 19644 genes. **K** Scatterplot of HER2E and LumA PAM50 centroid correlation coefficients. **L** Scatterplot of HER2E and LumB PAM50 centroid correlation coefficients. Statistical tests: Mann-Whitney U test + FDR correction (A-C); Mann-Whitney U test (E-I). Significance annotation: * ≤0.05; ** ≤0.01; *** ≤0.001; **** ≤0.0001. All reported p-values from statistical tests are two-sided. Boxplot elements correspond to: (i) center line = median, (ii) box limits = upper and lower quartiles, (iii) whiskers = 1.5x interquartile range. Source data are provided as a Source Data file.

To better understand *FGFR4* expression in our ERpHER2n tumors we generated matched DNA methylation profiles from 499 SCAN-B tumors using Illumina EPIC methylation beadchips (targeting approximately 800 K CpGs). Based on the clustering of beta-values (representing the level of DNA methylation) for CpGs mapping to the promoter region of *FGFR4* we identified a pattern of hypomethylation of CpGs annotated as shores in a subset of cases that showed matched elevated mRNA expression (Fig. 6A and Supplementary Fig. S6). In comparison, hypermethylation of these CpGs appear as the background state in normal breast tissue (Fig. 6B). Importantly, dividing the 499 SCAN-B cases by DNA methylation levels of two key CpGs (cg00618323 and cg20898288) in the shore region stratified tumors distinctly by *FGFR4* FKPM expression levels, with tumors showing hypomethylation of these CpGs having markedly elevated mRNA expression (Fig. 6C). Consistently, this division stratified also ERpHER2n HER2E tumors into a low and high *FGFR4* expression group, respectively (Fig. 6C). Consistent with Fig. 3I (i.e., presence of high *FGFR4* expressing tumors also in other PAM50 subtypes), we found that in all 499 tumors the CpG shore hypomethylated group, while enriched for HER2E tumors, comprised a mix of PAM50 subtypes (Fig. 6D).

To validate these findings and to extend them to breast cancer in general we performed a similar analysis in 645 TCGA breast cancers with matched DNA methylation and RNA-sequencing data. Again, a small subset of tumors with hypomethylation of the shore CpGs (only one shore CpG present in the lower resolution Illumina 450 K assay used in TCGA) were identified, which could define DNA methylation subgroups that were strongly correlating with *FGFR4* expression in a general breast cancer cohort (Fig. 6E, F). Similar to SCAN-B, while the hypomethylation cluster was enriched for HER2E tumors, both clusters comprised a mixture of PAM50 subtypes (Fig. 6G). The DNA methylation-based division of the TCGA tumors could also stratify HER2-positive tumors, irrespective of ER status and PAM50 HER2 status, into high and low *FGFR4* expression groups (Fig. 6H).

Finally, we also analyzed 48 human breast cancer cell lines with matched gene expression and DNA methylation data reported by Iorio et al.[28]. Similar to the SCAN-B and TCGA tumor cohorts, the DNA methylation status of the cg00618323 shore CpG was again strongly associated with *FGFR4* expression levels in the cell line cohort (Wilcoxon's *p* = 5e-05, Supplementary Fig. S6).

## Discussion

In the current study we performed a comprehensive clinicopathological and molecular characterization of ERpHER2n-HER2E (HER2E) breast cancer compared to the LumA and LumB subtypes but also compared to ERpHER2p tumors. Key objectives included investigating the HER2E subtype's clinical significance and gaining insight into characterizing molecular features that could be used to inform future treatment decisions.

Corroborating previous studies, HER2E tumors were associated with clinical features of aggressive disease akin to LumB such as high NHG and nodal positivity when compared to LumA tumors. As also reported previously, the HER2E subtype was associated with poor IDFS and OS in patients treated with endocrine therapy[11–15]. While a trend towards poor IDFS was also observed in HER2E patients treated with combined adjuvant chemotherapy and endocrine therapy, this observation requires validation in larger cohorts due to small patient numbers. Consistent with the generally poor outcome, 21-gene Recurrence Score and ROR score classification corroborated HER2E tumors to be a high-risk subgroup of ERpHER2n breast cancer patients. Agreement in risk classification (low/intermediate vs high risk) between commercial breast cancer risk signatures like OncotypeDX, Prosigna, and Mammaprint have been reported to be modest[29]. While the RNA-sequencing-based 21-gene Recurrence Score has not been validated versus the commercial implementation (OncotypeDX), the RNA-sequencing ROR scores for SCAN-B tumors have. In a pooled analysis of 148 ERpHER2n patients, an 86% agreement (kappa=0.72) between fresh frozen RNA-sequencing-based ROR risk groups (low/intermediate and high) and equivalent ROR risk groups based on matched FFPE tissue was reported[6], suggesting that HER2E tumors would typically be assigned a high-risk classification by commercial risk signatures. Together, patients with HER2E tumors represent a clinically important, albeit small (approx. 2-6%), high-risk subgroup of patients that should be identified and evaluated for new treatment approaches by in depth molecular characterization.

Analyses of *ERBB2* mRNA expression, IHC status, and gene amplification in the population-representative SCAN-B cohort demonstrated that HER2E tumors are not simply a group of misclassified HER2n cases (i.e., tumors that in reality are HER2p). This conclusion was further supported by the significantly lower *ERBB2* mRNA expression in HER2E tumors compared to HER2p tumors, and that HER2E tumors were not associated with a substantially elevated *ERBB2* mRNA expression compared to the other luminal subtypes. Moreover, our data show that the HER2E subtype is not equivalent to, or associated with, overrepresentation of the IHC-defined HER2-low group, and that HER2E tumors are not defined by frequent *ERBB2* mutations. The latter conclusion is corroborated by the MONALEESA study that observed no difference in the frequency of *ERBB2* mutations between PAM50 subtypes in hormone receptor-positive HER2-negative (HRpHER2n) metastatic disease[30]. Together, our observations about the role of *ERBB2* in HER2E tumors are in line with a recent report about the low impact of *ERBB2/GRB7* gene expression (both genes located within the 17q12 *ERBB2* locus) on the classification of the PAM50 HER2-enriched subtype[27].

The overall results of the molecular characterization of HER2E tumors on the RNA level indicate biological differences between the HER2E and the LumA subtypes but demonstrate that discriminating between HER2E and LumB tumors is substantially more challenging. This is likely partly due to that HER2E tumors exhibit highly proliferative characteristics similar to LumB tumors. The similarity to the LumB subtype was also reflected in PAM50 centroid correlation values of HER2E cases where the HER2E and LumB centroid values tended to be similar, and the LumB subtype typically represented the second-best PAM50 subtype fit for these cases as also noted by Veerla et al.[27]. Differential gene expression analysis identified a core set of 258 DEGs between the HER2E and LumA/LumB subtypes based on the SCAN-B and METABRIC cohorts. Interestingly, of these 258 genes none were differentially expressed between HER2E and HER2p-HER2E SCAN-B

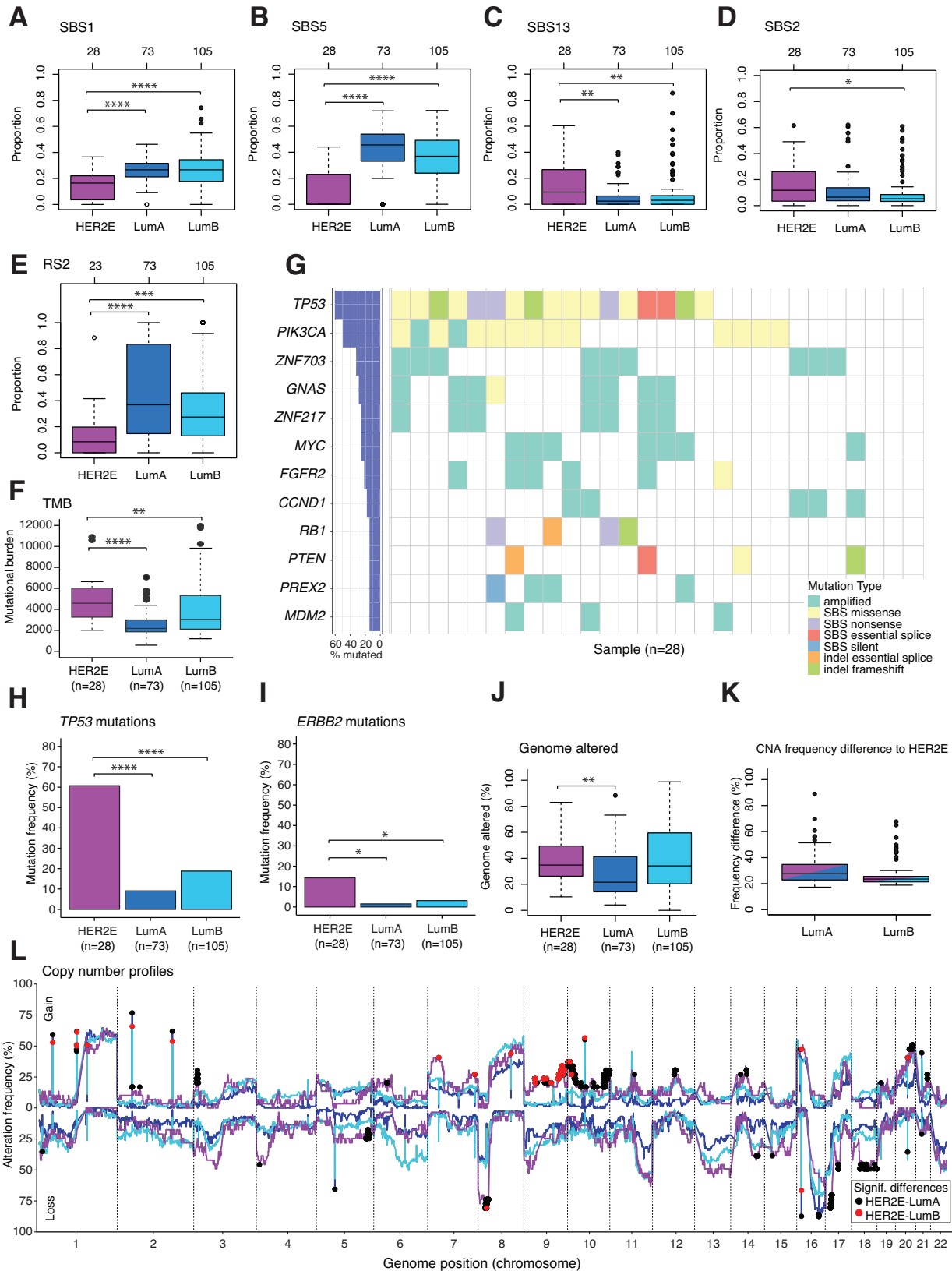

tumors, while 96% were differentially expressed between HER2E and HER2p-NonHER2E tumors. This finding reinforces that the transcriptional HER2E phenotype is not particularly dependent on the *ERBB2* gene status. The gene network analysis of this core DEG set identified the ETS-family GABPA transcription factor as enriched, which has been implicated in oncogenic mechanisms across multiple solid tumor

cancers and demonstrated to control cell migration in breast epithelial cells by regulating genes associated with cytoskeletal functions and cell migration control[31,32].

Other transcriptional hallmarks of HER2E tumors included a lower steroid response and a generally higher expression of immune response-associated genes (including the PD-L1 encoding gene *CD274*)

**Fig. 4 | PAM50 subtype-associated mutational and copy number alterations in the SCAN-B/BASIS patient set. A** SBS1 mutational signature proportions for tumors in respective subtype. **B** SBS5 mutational signature proportions for tumors in respective subtype. **C** SBS13 mutational signature proportions for tumors in respective subtype. **D** SBS2 mutational signature proportions for tumors in respective subtype. **E** RS2 rearrangement signature proportions for tumors in respective subtype (five HER2E cases excluded due less than 25 total rearrangements). **F** Tumor mutational burden (TMB) based on the sum of single base substitutions and indels per sample. **G** Mutational and amplification landscape of tumor drivers in HER2E tumors. **H** Mutation frequencies of *TP53*. **I** Mutation frequencies of *ERBB2*. **J** Proportion of tumor genomes affected by copy number alterations. **K** Differences in absolute alteration frequency of the statistically significant genes to HER2E. **L** Genome-wide copy number alteration profiles of gain and loss for HER2E (purple), LumA (blue), and LumB (light blue) tumors. Statistical tests: Mann-Whitney U test (A-D, F, I); Fisher's exact test (G-H); Fisher's exact test + FDR correction (K). Significance annotation: * ≤0.05; ** ≤0.01; *** ≤0.001; **** ≤0.0001. All reported p-values from statistical tests are two-sided. Boxplot elements correspond to: (i) center line = median, (ii) box limits = upper and lower quartiles, (iii) whiskers = 1.5x interquartile range. Top-axis in boxplots reports group sizes. Source data are provided as a Source Data file.

compared to LumA and LumB tumors. The latter suggests a more immune cell infiltrated tumor microenvironment in HER2E tumors, as also noted by Griguolo et al.[33] This was further supported by the analyses of phenotyped single-cell data obtained from Danenberg et al.[23] that highlighted HER2E tumors to exhibit a trend towards a higher enrichment of immune cells (CD4 + T-cells, CD8 + T-cells, macrophages, and B-cells) compared to LumA and LumB tumors. It should be acknowledged that the single-cell analysis was based on comparisons across clinical groups, necessitated by the low number of HER2E subtyped ERpHER2n cases in the Danenberg et al. study. The lower steroid response (including lower *ESR1* expression) in HER2E tumors may serve as a possible explanation for the poor response to endocrine therapy and are consistent with a hypothesis that acquisition of the HER2E subtype in metastatic disease is linked to estrogen independence[16], further supporting chemotherapy as part of early management for patients with HER2E tumors. Consistent with the metastatic HRpHER2n-focused MONALEESA clinical trial[30], we did not find evidence of difference in *ESR1* mutation frequency between the PAM50 subtypes in our primary breast cancer cohorts that may explain the lower *ESR1* expression and the obviously poorer response to endocrine therapy in HER2E tumors.

On an individual gene level our mutational characterization identified a high frequency of *TP53* mutations as a distinct feature of the HER2E subtype compared to the other luminal subtypes, while similar to the frequency in HER2p tumors. This observation is consistent with the general association of *TP53* mutations with a poorer outcome in luminal disease[34–36] and with the reported observation of a comparatively higher frequency of *TP53* alterations in HER2E tumors also in HRpHER2n metastatic disease[30]. Regarding broader mutational patterns defined by mutational signature exposures, HER2E tumors demonstrated higher HRD frequency and higher exposure to the APOBEC-related SBS2 and SBS13 mutational signatures, in line with reports of higher APOBEC activity in HER2-amplified/HER2-enriched tumors[37,38]. Consistent with higher APOBEC exposure, the frequency of kataegis (a local hypermutation phenomenon associated with APOBEC[24]) was also higher in HER2E tumors compared to the other luminal subtypes.

Kataegis loci frequently co-localize with specific rearrangement signatures at the vicinity of structural rearrangements[24,26,39]. In this context, breast cancer has been proposed to be a malignancy that is substantially governed by copy number alterations (CNAs) rather than being exclusively influenced by oncogenic mutations[40]. Our copy number-based characterization of ERpHER2n tumors demonstrated a higher overall burden of CNAs in HER2E and LumB compared to LumA tumors. Again, this finding agrees with the perception of generally more aggressive disease in LumB and HER2E subtyped tumors. The genome-wide copy number analysis further demonstrated the dissimilarity between the HER2E subtype and the LumA subtype, while also showing that on a global copy number level the HER2E and LumB subtypes do not appear substantially different. Together, the genome-wide analyses did not identify regions of CNAs that can be viewed as defining traits for the HER2E subtype in luminal disease.

While this study leverages the two largest single-institution cohorts of primary breast cancer with substantial associated

molecular data reported to date there are still limitations. For instance, the study is limited by low numbers of HER2E tumors and mixed patient treatments (e.g., different endocrine therapies) due to either potential patient selection bias (METABRIC) or low population frequency (SCAN-B). The observed discrepancy in HER2E prevalence between the SCAN-B and METABRIC cohort is possibly explained by the different definitions of ER-positivity (≥10% in SCAN-B and expected ≥1% in METABRIC) and the SCAN-B cohort being more population representative. Due to specific ER% not being available, we could not reclassify METABRIC cases according to Swedish national guidelines to determine if the ER cut-off level was the main source of the frequency bias between the cohorts. Important to note is that neither SCAN-B nor METABRIC are randomized clinical trials. As SCAN-B is a population-representative observational study it reflects national treatment guidelines during the inclusion period, grounding the findings in real-world oncology management practices. Patients do not randomly receive a specific treatment, meaning that there are potential competing risks due to age, patient preferences, and comorbidities that can influence survival analyses. Moreover, modern palliative oncological treatment (relevant for SCAN-B patients) can support patients for a considerable time. Combined, these aspects have implications for how overall survival as an endpoint can be interpreted. Other study limitations include the limited mutational data available for METABRIC that hindered deeper sequencing analyses in this cohort, the low number of WGS-analyzed HER2E SCAN-B cases that affects statistical significance, and the lack of matched in situ cell type estimates for delineation of the tumor microenvironment. Finally, while the current study has carefully characterized HER2E tumors on the clinicopathological, RNA, and DNA levels, it does not address whether HER2E tumors have distinct epigenetic alterations compared to LumB and LumA tumors.

Considering the aggressive nature of HER2E tumors there is an imminent need to expand current treatment options for HER2E patients beyond chemotherapy. One possible alternative treatment option for HER2E tumors is cyclin-dependent kinase 4/6 (CDK4/6) inhibitors today administered in combination with an endocrine backbone[41]. Being the standard of care in the first-line metastatic setting of HRpHER2n metastatic breast cancer, CDK4/6 inhibitors represent a promising therapeutic approach to improving the care of patients in the early setting as well[42]. Notably, Prat et al. demonstrated improved progression-free survival of patients with HER2E, LumA, and LumB subtyped metastatic HRpHER2n tumors included in the MONALEESA trials, which diverges from preclinical data that indicates non-luminal subtypes not benefitting much from CDK4/6 inhibition[10,13]. Moreover, two recent clinical trials demonstrated that addition of the CDK4/6 inhibitors Abemaciclib (MonarchE trial) or Ribociclib (NATALEE trial) to endocrine therapy significantly improved invasive disease-free survival also in the high-risk HRpHER2n early setting[43,44].

Another alternative treatment option based on the observation of a generally more immune active tumor microenvironment in HER2E tumors is immunotherapy. Immune checkpoint inhibitors (ICIs) have recently become clinical routine as part of neoadjuvant therapy in TNBC based on e.g., the Keynote-522 clinical trial[45]. The preliminary results of two currently ongoing clinical trials assessing the

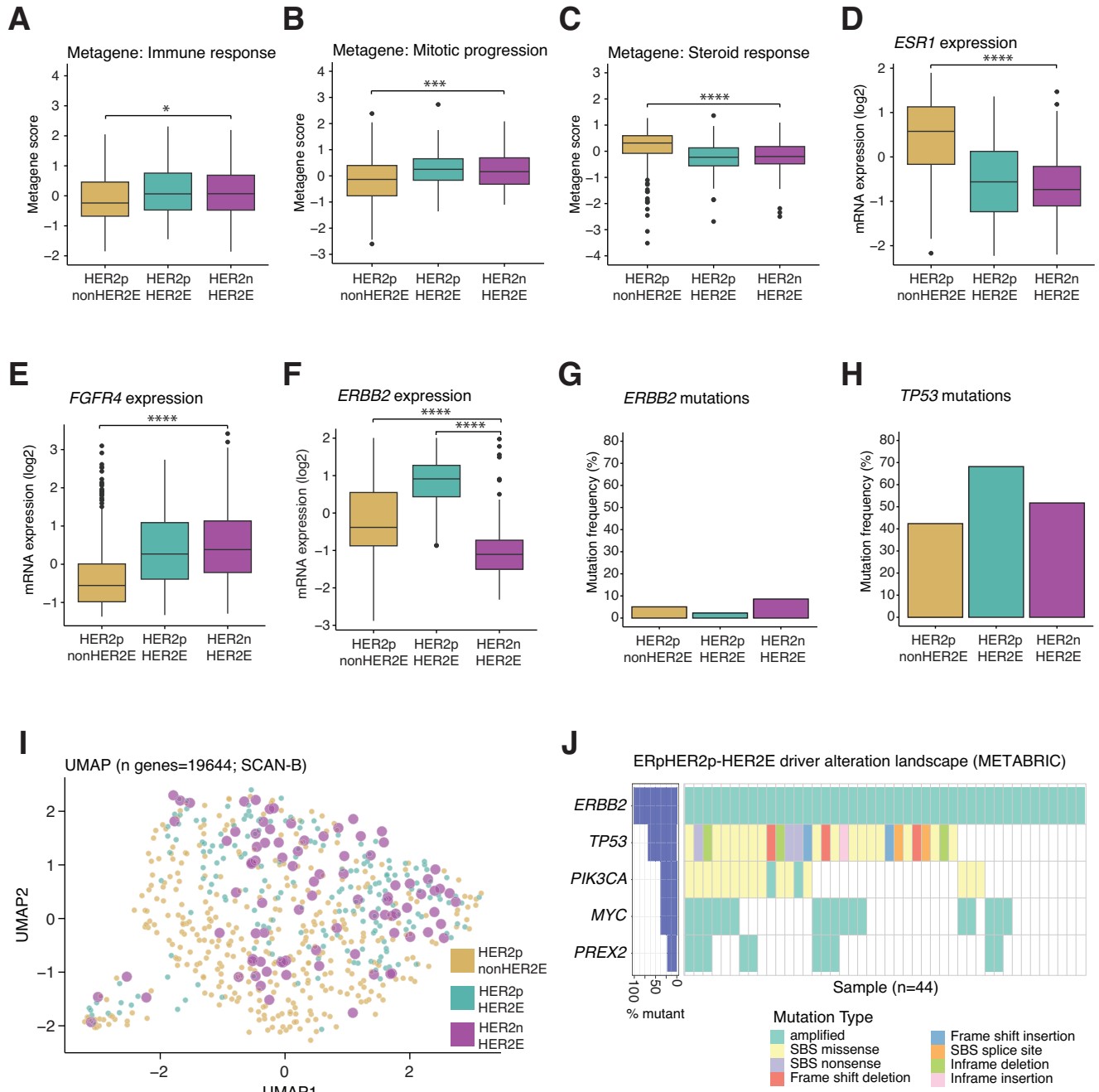

**Fig. 5 | Molecular feature comparisons between HER2E and HER2p-HER2E and HER2p-NonHER2E.** Analyses using SCAN-B tumors were based on 89 HER2E tumors, 193 HER2p-HER2E tumors, and 371 HER2p-nonHER2E tumors. **A** Immune response metagene scores. **B** Mitotic progression metagene scores. **C** Steroid response metagene scores. **D** Scaled mRNA expression of *ESR1*. **E** Scaled mRNA expression of *FGFR4*. **F** Scaled mRNA expression of *ERBB2*. **G** Mutation frequencies of *ERBB2* excluding amplifications. **H** Mutation frequencies of *TP53*. **I** UMAP analysis based on FPKM expression data of 19644 genes. **J** Mutational and amplification landscape of tumor drivers in HER2p-HER2E tumors. Statistical tests: Mann-Whitney U test + FDR correction (A-C); Mann-Whitney U test (D-F); Fisher's exact test (G-H). Significance annotation: * ≤0.05; ** ≤0.01; *** ≤0.001; **** ≤0.0001. All reported p-values from statistical tests are two-sided. Boxplot elements correspond to: (i) center line = median, (ii) box limits = upper and lower quartiles, (iii) whiskers = 1.5x interquartile range. Source data are provided as a Source Data file.

combination of a PD1 inhibitor (Pembrolizumab in KeyNote-756 and Nivolumab in CheckMate-7FL) with neoadjuvant chemotherapy in high-risk ERpHER2n early breast cancer demonstrated significant improvement in pathological complete response yet awaiting event-free survival data[46,47]. As a higher immune response has been associated with a favorable prognosis in *HER2*-amplified disease and it is a positive biomarker for response to ICIs in metastatic TNBC, further studies are pertinent to assess whether the PAM50 HER2E subtype can be used as a surrogate marker for luminal tumors that might have a

higher response rate to ICIs[48,49]. If so, the HER2E subtype would have clinical relevance within luminal breast cancer, a subgroup that in general has been reported to generally respond more poorly to ICIs compared to TNBC, likely due to a generally colder immune environment[50].

Therapy options indicated for subsets of HER2E patients are PARP inhibitors and other HRD-directed therapeutics based on positive HRD status, and antibody-drug-conjugates (ADCs) based on HER2-low status. Regarding PARP inhibitors, the OlympiA trial was able to

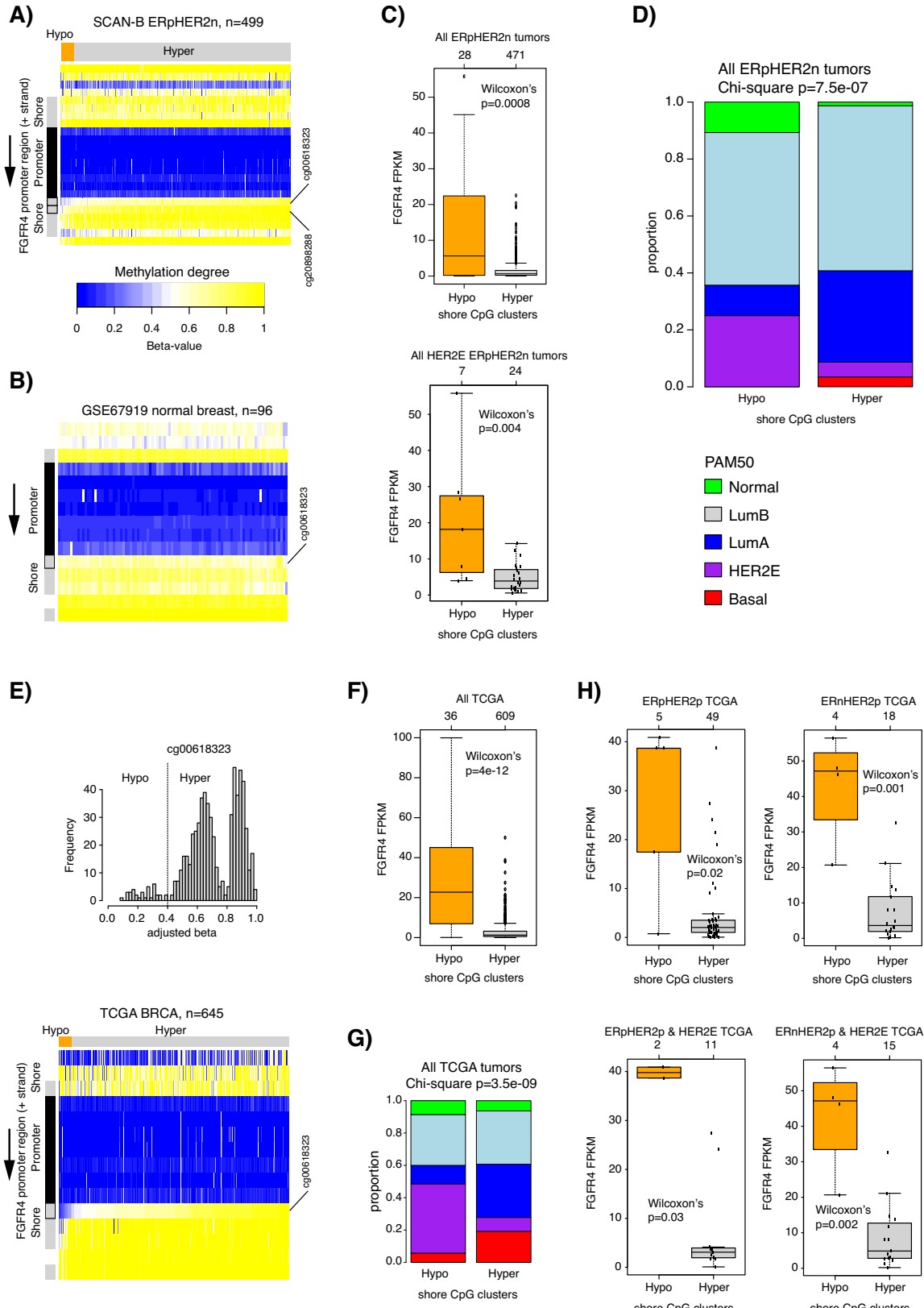

demonstrate an improved invasive disease-free survival among patients with high-risk, HER2-negative early breast cancer and germline *BRCA1/2* pathogenic variants receiving an adjuvant PARP inhibitor (Olaparib)[51]. HER2E tumors indicated as HRD-positive may also benefit from a combination of neo-adjuvant chemotherapy and a PARP inhibitor, considering results from the GeparOLA study within early HER2-

negative HRD-positive breast cancer, where patients with hormone receptor-positive tumors exhibited higher response rates[52,53]. ADCs are currently receiving high attention in breast cancer, with two drugs recently approved by the FDA and EMA for patients with endocrine therapy-resistant HRpHER2n metastatic disease. The HER2-targeting ADC Trastuzumab Deruxtecan was proven to benefit patients with

**Fig. 6 | Regulation of *FGFR4* expression by promoter shore hypomethylation in breast cancer. A** DNA methylation beta values for CpGs associated with *FGFR4* for 499 SCAN-B ERpHER2n tumors analyzed by Illumina EPIC beadchips. Black=Promotor, Gray=shore. Tumors are grouped by beta-values for the two shore CpGs cg00618323 and cg20898288, with a cut-off of 0.6. **B** Beta values for *FGFR4* CpGs in 96 normal breast tissue specimens from GSE67919. **C** Top panel: *FGFR4* FPKM expression in the 499 SCAN-B tumors grouped according to shore CpGs. Bottom panel: *FGFR4* FPKM expression in ERpHER2n HER2E SCAN-B tumors grouped according to shore CpGs. **D** Distribution of PAM50 subtypes in the SCAN-B tumors by CpG status. **E** Top panel, histogram of beta values for cg00618323 in 645 TCGA tumors analyzed by Illumina 450 K beadchips. cg20898288 is not present on this array. A cut-off of 0.4 was used define hypomethylation and hypermethylation as

the TCGA data was corrected for tumor cell content, thereby removing a considerable impact of normal cell contamination as seen in the beta histogram. Bottom panel: corresponding CpG beta value heatmap for the 645 TCGA tumors. **F** *FGFR4* FPKM expression in the 645 TCGA tumors grouped according to shore CpGs. **G** Distribution of PAM50 subtypes in the TCGA tumors by CpG status. **H** *FGFR4* FPKM expression in HER2-positive tumor subgroups in the TCGA cohort defined by ER-status and HER2E status according to shore CpGs. All reported p-values from statistical tests are two-sided. Boxplot elements correspond to: (i) center line = median, (ii) box limits = upper and lower quartiles, (iii) whiskers = 1.5x interquartile range. Top-axis in boxplots reports group sizes. Source data are provided as a Source Data file.

HER2-low metastatic disease in the DESTINY-Breast04 trial[54]. Thus, while our analyses of HER2-low frequency indicate that HER2-targeted ADCs should not be considered as baseline therapy for HER2E tumors, the HER2-low subset within HER2E tumors may still benefit. However, HER2-zero tumors may also benefit from ADCs, for example in form of TROP2-targeted agents such as Sacituzumab Govitecan, which is effective in treating metastatic HRpHER2n breast cancer as demonstrated by the TROPiCS-02 trial[55].

Finally, a promising therapeutic approach on a more group level for the PAM50 HER2-enriched subtype may be based on inhibiting *FGFR4*. Selected for being prototypically overexpressed in HER2-enriched tumors[4], high *FGFR4* (located on 5q35.2) expression represents a typical defining trait irrespective of clinical ER and HER2-status. Consistently, *FGFR4* was identified in our differential gene expression analysis between HER2E and LumA/LumB tumors, as well as versus ERpHER2p-NonHER2E tumors. In general, fibroblast growth factor receptors like *FGFR4* are involved in the development, differentiation, cell survival, migration, angiogenesis, and carcinogenesis[56] by intracellular signal transduction through e.g., the RAS/RAF/MEK and PIK3K/AKT pathways[57]. *FGFR4* activation has been associated with cancer progression and resistance to different types of anti-cancer therapy and preclinical studies have shown that FGFR4 knockdown or pharmacologic inhibition can inhibit tumor growth and metastasis both in vitro and in vivo (see[58] for review). Consistent with the latter, the recent study by Garcia-Recio et al. demonstrated tumor growth inhibition in a HER2-negative HER2E PDX-model treated with an FGFR4 inhibitor, but also molecular subtype switching in cancer cells due to *FGFR4* inhibition[59]. Multiple strategies of inhibiting *FGFR4* have been developed including small molecule inhibitors, neutral antibodies, and an extracellular protein trap being tested in ongoing clinical trials in different malignancies[58]. Considering the preclinical data presented in breast cancer, investigating the vulnerability of primary HER2E tumors specifically to *FGFR4* inhibition appears relevant. However, while *FGFR4* expression is selected to be a defining PAM50 trait for HER2E tumors expression is still heterogeneous, including both high expressing tumors of other subtypes but also variability in expression within the HER2E subtype, irrespective of *ERBB2* amplification status. This simple illustration warrants further investigation of other genes selected to be prototypical drivers of specific PAM50 subtypes. In this study we demonstrate that the regulator of *FGFR4* mRNA expression are not somatic variants or copy number alterations but instead appears to be DNA hypomethylation of regions in the gene's promoter, acting as an apparent switch for *FGFR4* expression in breast cancer. In more detail, it explains heterogeneity in *FGFR4* expression both across PAM50 subtypes and in cancer cell lines, but also within HER2E tumors irrespective of *ERBB2* status, and may even be potentially informative as a biomarker for FGFR4 inhibitors.

In conclusion, our analyses demonstrated that HER2E tumors in ERpHER2n early breast cancer are neither driven by *ERBB2* alterations nor predominantly constituted by HER2-low tumors. Molecular features that distinctly characterize the HER2E subtype in luminal disease were shown to be consistent with the HER2-enriched subtype in HER2-

positive disease, including a putative epigenetic mechanism for *FGFR4* expression in primary breast cancer tissue as well as in cell lines. Lastly, we highlighted several new treatment possibilities based on potentially targetable molecular features that may offer future alternatives for patients with this aggressive tumor phenotype.

## Methods

### Inclusion and Ethics statement

Patients were enrolled in the Sweden Cancerome Analysis Network – Breast (SCAN-B) study (ClinicalTrials.gov ID NCT02306096)[60,61] approved by the Regional Ethical Review Board in Lund, Sweden (registration numbers 2009/658, 2010/383, 2012/58, 2013/459, 2014/521, 2015/277, 2016/541, 2016/742, 2016/944, 2018/267 and the Swedish Ethical Review Authority (registration numbers 2019-01252, 2024-02040-02), governed by the Swedish Ethical Review Authority, Box 2110, 750 02 Uppsala, Sweden. All patients provided written informed consent prior to enrolment. All analyses were performed in accordance with patient consent and ethical regulations and decisions.

### Patient cohorts

The analyses in this study were based on the early-stage breast cancer datasets from two large independent cohorts, the Sweden Cancerome Analysis Network – Breast (SCAN-B) study[60,61] and The Molecular Taxonomy of Breast Cancer International Consortium (METABRIC) study[21]. Cases from the population-representative SCAN-B cohort recently reported by Staaf et al.[6] constituted the focal point of analyses included in the clinicopathological and molecular characterization of the HER2E subtype. Based on the cohort reported by Staaf et al.[6] we extracted clinical cancer registry data and RNA-sequencing data from 4487 ERpHER2n cases. Of these 4487 tumors, 89 (2%) were classified as HER2E, 3049 (68%) as LumA, and 1349 (30%) as LumB according to available PAM50 nearest centroid classification. Clinical cancer registry data was complemented with a clinicopathological review of patients' medical charts for 904 patients focusing on updated relapse status, clinical HER2-status (IHC 0, IHC 1 + , or IHC 2 + /ISH-) used to create a HER2-low classification, and clinical variables (lymph node status, Nottingham histological grade (NHG), and tumor size) to be used as covariates in multivariate survival analysis. Expression data for 19675 genes was available as FPKM (Fragments Per Kilobase per Million mapped fragments) estimates from the online repository associated with the study by Staaf et al.[6]. Whole genome sequencing (WGS) was performed for 28 of the 89 HER2E-classified tumors as outlined below.

The METABRIC cohort provided clinical, mutational (targeted NGS), transcriptomic (Illumina HT-12 v3 platform), and copy number (Affymetrix SNP 6.0 platform) data for 999 ERpHER2n cases with available PAM50 class[21,62]. Of the 999 cases, 58 (6%) were PAM50 subtyped as HER2E, 601 (60%) as LumA, and 340 (34%) as LumB. METABRIC data was downloaded from the cBioPortal website (https://www.cbioportal.org/) as pre-compiled data. Expression data for 24368 genes were available as z-score estimates. In contrast to the used SCAN-B cohort, a comprehensive population representativity

analysis for the METABRIC cohort, drawn from a cohort of patients diagnosed between 1977 and 2005[63], has not been reported.

A third smaller cohort was used for comparison to the 28 SCAN-B WGS cases. We obtained WGS and RNA-sequencing data (log2 transformed FPKM data) from 178 ERpHER2n cases, including 73 LumA and 105 LumB classified tumors from the study by Nik-Zainal et al.[26], hereafter referred to as BASIS.

Patient inclusion and exclusion criteria for the used cohorts are available from original publications. In all cohorts, the PAM50 subtype characterization was obtained from deposited data and used to form the class labels for the subsequent analyses. In the METABRIC cohort the HER2-status was determined by using the SNP 6.0 copy number array data for the *ERBB2* gene, whereas in BASIS it was verified from WGS data for *ERBB2*. In the SCAN-B cohort HER2-status was determined from routine clinical IHC and FISH analysis as reported in the Swedish National Breast Cancer Quality Registry (NKBC). In SCAN-B, ER-positivity was defined as ≥10% of tumor cells being IHC-stained in line with Swedish national guidelines. ER and PR data is based on routine clinical IHC and FISH analysis as reported in the NKBC registry. In the METABRIC cohort ER-positivity is expected to be defined as ≥1% IHC-stained tumor cells. Due to specific ER IHC scores not being available, reclassifying METABRIC cases according to Swedish national guidelines was not possible. In all analyses, a tumor sample is represented by a single analysis (e.g. WGS analysis, RNA-sequencing, DNA methylation analysis).

## DNA methylation analysis of SCAN-B ERpHER2n tumors
DNA methylation analysis was performed on the same DNA used for WGS using Illumina EPIC v1 beadchips according to manufacturer's instructions for 499 tumors. Methylation profiling was performed by the SNP&SEQ Technology Platform in Uppsala (www.genotyping.se). Beta values, representing the level of methylation, were computed in a sample-by-sample context using the minfi R package v1.44 function preprocessNoob() and Infinium probe normalized using the approach described by ref. 64.

## The Cancer Genome Atlas (TCGA), normal breast tissue, and breast cancer cell line cohorts
Processed and compiled DNA methylation data (Illumina 450 K beadchips) and RNA-sequencing FPKM estimates for 645 breast cancers with matched data from the TCGA cohort was obtained from the study by Staaf and Aine[65]. DNA methylation data (beta-values) were adjusted for tumor cell content as described in ref. 65 for each tumor. TCGA tumors were PAM50 subtyped using the NCN approach described by Staaf et al.[6], using the same reference sets and code. DNA methylation profiles for 96 normal breast tissue samples analyzed by the Illumina 450 K methylation platform were obtained from GSE67919 as processed beta-values[66]. Gene expression data and matched DNA methylation profiles for 48 breast cancer cell lines analyzed by Affymetrix RNA microarrays and Illumina 450 K methylation arrays were obtained from the study by Iorio et al.[28] as processed RMA expression values and beta-values (GSE68379), respectively.

## Statistics and survival analyses
Statistical methods are further detailed in the Supplementary Methods. Comparisons of clinicopathological variables were performed using Fisher's exact test and parametric tests. Reported p-values are two-sided. Survival analyses were performed in R (v4.2.2) using the survival (v3.4.0) and survminer (v0.4.9) packages. Survival curves were estimated using the Kaplan-Meier method and compared using log-rank tests. Hazard ratios were calculated through univariable and multivariable Cox regression using the coxph R function. Invasive disease-free survival (IDFS, in SCAN-B), recurrence-free interval (RFI, in METABRIC) and overall survival (OS, in both SCAN-B and METABRIC) were used as clinical endpoints. Multivariate Cox's proportional

hazards models included patient age at diagnosis, lymph node status, tumor size, and tumor grade as covariates to PAM50 subtypes. In SCAN-B, covariates for multivariate survival analysis were taken from the clinical review data for matched samples. Cases were grouped based on whether they received only endocrine therapy (ET) or a combination of adjuvant chemotherapy and endocrine therapy (CT + ET). The median outcome measures in censored patients were equal to 10.3 years (RFI, ET, METABRIC), 9.7 years (RFI, CT + ET, METABRIC), 11.1 years (OS, ET, METABRIC), 10.8 years (OS, CT + ET, METABRIC), 9.9 years (IDFS, ET, SCAN-B), 8.6 years (IDFS, CT + ET, SCAN-B), 7.1 years (OS, ET, SCAN-B), and 6.7 years (OS, CT + ET, SCAN-B). Most SCAN-B patients receiving CT were treated with a combination of FEC (Fluorouracil, Epirubicin, and Cyclophosphamide) and Docetaxel. Patients receiving ET were treated with tamoxifen and/or aromatase inhibitors. Detailed treatment information was not available for the METABRIC and BASIS cohorts.

## Gene expression analyses
The transcriptomic characterization of ERpHER2n-HER2E breast cancer included expression analyses of selected genes and metagenes, supervised differential gene expression analysis, pathway enrichment, transcription factor binding analysis, both supervised and unsupervised sample clustering, and comparisons of PAM50 centroid correlations. For all gene expression analyses, the SCAN-B cohort data preprocessing consisted of log2-transforming the FPKM gene expression data with a + 1 offset and subsequently gene-wise scaling by applying a z-transformation. Deposited METABRIC data was already preprocessed (including z-transformation) and therefore directly used. Gene expression scores for six reported biological metagenes related to breast cancer biology (termed basal, lipid, mitotic progression, immune response, steroid response, and stroma)[22] were calculated for each sample as the average of all associated gene expression values. For metagene comparisons, false-discovery rate (FDR) was used to correct for multiple testing by employing the p.adjust function of the stats R package (v3.6.2).

In silico abundance estimates of different cell types were obtained for all samples from Nacer et al.[67] derived from bulk RNA-sequencing data by the CIBERSORTx deconvolution-based method[68]. CIBERSORTx results were filtered to keep only samples with $p \leq 0.05$ ($n = 4377$ kept samples) since lower p-values are connected to a more reliable deconvolution process.

Differentially expressed genes between ERpHER2n breast cancers PAM50 classified as HER2E and LumA or LumB were identified by two-sided t-tests in combination with Bonferroni multiple testing correction. Genes with an adjusted $p$ value $\leq 0.05$ were considered as differentially expressed. The core sets of differentially expressed genes (DEGs) were defined by genes that were differentially expressed both in the SCAN-B and METABRIC cohorts. Thereby, three core sets were defined: 1) genes differentially expressed in both HER2E vs. LumA and HER2E vs. LumB (HER2E cDEGs), 2) genes differentially expressed in HER2E vs. LumA (LumA cDEGs), and 3) genes differentially expressed in HER2E vs. LumB (LumB cDEGs). Core sets of identified DEGs were investigated for enriched pathways using the enrichR package (v3.2) accessing the WikiPathways database[69,70]. Transcription factor binding, Risk of Recurrence score, and 21-gene recurrence score (RS) analyses were performed as described in Supplementary Methods. Unsupervised UMAP analysis was performed to investigate the presence of subtype-associated gene expression patterns by using the umap R package (v0.2.10.0).

## Analysis of Immune Cell Proportions using Single-Cell Phenotype data
The specific cellular composition within HER2E tumors was investigated by using matched in situ single-cell estimates of lymphocyte counts derived from single-cell METABRIC data ($n = 612$ patients), as

presented by Danenberg et al.[23]. Using this data, we quantified the counts of different cell phenotypes within each sample, which were then used to calculate the proportions of immune cell types. We performed a comparative analysis of these immune cell counts and proportions across different PAM50 breast cancer subtypes, focusing on identifying patterns in immune cell infiltration.

## Whole genome sequencing of SCAN-B cases

WGS to an average depth of 37X was performed on tumors and matched normal DNA samples for 28 ERpHER2n HER2E SCAN-B cases by using Illumina paired-end sequencing on S4 flow cells with NovaSeq equipment performed at Novogene UK (www.novogene.com). Analysis of somatic alterations (single nucleotide variants, SNVs, indels, rearrangements), copy number profiles, were performed as described previously[26]. Single base pair substitution and structural rearrangement mutational signatures were performed as outlined[71] and described in Supplementary Methods. Homologous recombination deficiency (HRD) was assessed using HRDetect[25]. Analysis of kataegis was performed on WGS data from SCAN-B and BASIS tumors using the KataegisPortal R package v1.0.3[72] using default settings. Only kataegis events with confidence ≥1 were kept for further analyses. A binary kataegis positive status was inferred for each tumor if ≥1 event was recorded.

## Copy number alteration analyses

Copy number analyses were performed in the METABRIC cohort using Affymetrix SNP6 array data ($n = 999$ cases in total of which $n = 58$ classified as HER2E) and in the SCAN-B cohort using copy number estimates obtained from WGS ($n = 28$ HER2E cases). For WGS-based SCAN-B data, we compared HER2E profiles to copy number estimates from ERpHER2n LumA ($n = 73$) and LumB ($n = 105$) tumors from the BASIS cohort. WGS data was processed for copy number analyses as described in ref. 73 and Supplementary Methods, including allele-specific segmentation using ASCAT and calling of copy number alterations versus tumor ploidy. In METABRIC, deposited copy number data for 22544 genes provided a copy number state of neutral (0), gain (1), amplification (2), loss (-1), or low-deletion/homozygous deletion (-2). Further details about copy number analysis are found in the Supplementary Methods.

## Comparison of ERpHER2n-HER2E to ERpHER2p breast cancer

We further investigated distinguishing features of HER2E cases in the ERpHER2n subgroup by comparing them with cases that were clinically ER-positive/HER2-positive (ERpHER2p). Comparisons of transcriptomic features were done using SCAN-B data, whereas mutational features were compared using METABRIC data (both processed as previously described). For these comparisons, the two contrasting groups within ERpHER2p cases were defined as cases that were PAM50 subtyped as HER2-enriched (HER2p-HER2E) or another subtype (HER2p-NonHER2E). The SCAN-B cohort provided data on 564 cases (HER2p-HER2E $n = 193$; HER2p-NonHER2E $n = 371$) and the METABRIC cohort on 162 cases (HER2p-HER2E $n = 44$; HER2p-NonHER2E $n = 118$).

## Data availability

The raw whole genome sequencing data for SCAN-B cases are protected and are not available due to data privacy laws. The processed somatic whole genome sequence tumor data are available in Supplementary Dataset 4. The DNA methylation data generated in this study have been deposited in the Gene Expression Omnibus database under accession code GSE278586. DNA methylation data for the *FGFR4* locus for the different cohorts (SCAN-B and TCGA) are included in Supplementary Dataset 5. The SCAN-B RNA-sequencing data used in this study are available from Staaf et al.[6] [https://data.mendeley.com/datasets/yzxtxn4nmd/3]. The breast cancer cell line data used in this

study are available from [https://www.ncbi.nlm.nih.gov/geo/query/acc.cgi?acc=GSE68379]. The TCGA data used in this study is available from the GDC data portal [https://portal.gdc.cancer.gov]. The single cell estimates of lymphocyte counts from Danenberg et al. used in this study is available from a Zenodo repository [https://zenodo.org/records/5850952]. The normal breast tissue DNA methylation data used in this study are available from [https://www.ncbi.nlm.nih.gov/geo/query/acc.cgi?acc=GSE67919]. The METABRIC data used in this study are available from the cBioPortal website (https://www.cbioportal.org/). Raw data for the BASIS cohort used in this study are available from the EGA repository [https://ega-archive.org/studies/EGAS00001001178]. Source data are provided with this paper. Source data are provided with this paper.

## Code availability

All analyses and associated statistical tests were performed using the R programming language (v4.0.3). The scripts for all analyses that were performed as part of this study are freely accessible at: https://github.com/StaafLab/HER2E_NatCom[74].

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

## Acknowledgements

The authors would like to acknowledge patients and clinicians participating in the SCAN-B study, personnel at the central SCAN-B laboratory at the Division of Oncology, Lund University, the Swedish National Breast Cancer Quality Registry (NKBC), Regional Cancer Center South, RBC Syd, and the South Sweden Breast Cancer Group (SSBCG). Methylation profiling was performed by the SNP&SEQ Technology Platform in Uppsala (www.genotyping.se). The facility is part of the National Genomics Infrastructure supported by the Swedish Research Council for Infrastructures and Science for Life Laboratory, Sweden. Financial support for this study was provided by the Swedish Cancer Society (CAN 2021/1407 JS, 2024/3591 JS), the Mrs Berta Kamprad Foundation (FBKS-2020-5 JS, FBKS-2024-14 JS), the Swedish Research Council (2021-01800 JS), the Swedish Breast Cancer Association (JS), the Mats Paulsson Foundation (ÅB), the Cancera Foundation (ÅB), and Swedish governmental funding (ALF, grant 2022/0021 JS). The funders had no role in study design, data collection and analysis, decision to publish, or preparation of the paper.

## Author contributions

Conception and design: L.H., J.S. Collection and assembly of data: L.H., J.S., J.V.C., A.B.C., K.S., J.H., F.R. Provision of study material or patients: J.V.C., Å.B. Data analysis and interpretation: L.H., J.C., D.F.N., J.H., H.D., S.N.Z., Y.M., D.B., R.B., S.V., N.N., P.T.R. Financial support: J.S., Å.B. Administrative support: J.V.C. Manuscript writing: L.H., J.S. with input from all authors. Final approval of manuscript: All authors. Agree to be accountable for all aspects of the work: All authors.

## Funding

## Competing interests

S.N.-Z. and H.R.D. hold patents or have submitted applications on clinical algorithms of mutational signatures: HRDetect (PCT/EP2017/060294), clinical use of signatures (PCT/EP2017/060289) and clinical predictor (PCT/EP2017/060298). S.N.Z. also holds the following patents; MMRDetect (PCT/EP2022/057387), rearrangement signature methods (PCT/EP2017/060279) and hotspots for chromosomal rearrangements (PCT/EP2017/060298). Two further patent filings have been made recently (numbers are pending). All other authors declare that they have no competing interests.
