## [Transparent Peer Review file · Nature Communications]

Genomic characterization of the HER2-enriched intrinsic molecular subtype in primary ER-positive HER2-negative breast cancer

Corresponding Author: Dr Johan Staaf

Version 0:

Reviewer comments:

Reviewer #1

(Remarks to the Author)

This manuscript focuses on clinicopathological and molecular characterization of a small subset of HER2 enriched tumors that were classified as ER-positive /HER2-negative (ERpHER2n) based upon ER status but, when classified by the PAM50 molecular signature, show up as a small subset of ERp tumors (termed HER2E). These are not considered ERpHER2p or just ERp but rather have an in-between status. The key goal of this paper is to distinguish these tumors from those that are luminal A or B ERp or technically classified as HER2p rather than HER2 enriched using two main datasets, and to use this information to guide future treatment. While the authors do find some statistically significant differences in aspects of the molecular features of this tumor subset, the concern that arises is the clinical significance of these differences particularly as a guide to distinguish this group from other tumors such as luminal B for treatment. The main limitation of this analysis is the bioinformatic reliance on archived data without supplementation using experiment-based validation. Overall, the general conclusions are relatively minimal, citing the immune response and FGFR4 as potentially targetable features without other supporting evidence.

Specific comments:

- 1) As noted by the authors, the aggressive nature of ERpHER2p tumors has been reported previously for patients treated with endocrine therapy. However, their attempts to extend this to patients treated with chemotherapy for overall survival, as they note, was not significant for the SCAN-B dataset but rather showed "a trend towards poor IDFS ... this observation requires validation in larger cohorts due to small patient numbers." Similarly, the RFI analysis of patients from the METABRIC dataset treated with chemotherapy (CT) and endocrine therapy (ET) is technically significant by p value but limited to 4 HER2E patients out of a total of 72 total. Furthermore, HER2E patients treated with CT and ET did not have significantly different overall survival intervals relative to other luminal subtypes, so the clinical significance is not clear.
- 2) Many of the observations (e.g. frequency of HER2-low samples) are inconsistent between the two tumor datasets so it is difficult to determine the clinical significance.
- 3) Molecular features of ERpHER2p like RNA expression and copy number showed biological differences with Luminal A but were not readily distinguishable from luminal B patients. The networks they identified were not enriched for any pathways and little insight could be gained from transcription factor analysis.
- 4) The focus on the immune system was, as noted, limited by inferring immune infiltration using CIPERSORTx rather than direct demonstration.

Reviewer #2

(Remarks to the Author)

Comments:

This paper looks at genomic alterations (both DNA and RNA) in a small but distinct subtype (ER positive/HER 2 negative clinically but HER-2 enriched by PAM50) relative to luminal A and luminal B subtypes in two previously published separate

cohorts [SCAN-B and METABRIC] of early stage breast cancer treated with older (and not necessarily specified) systemic treatments. The authors have further characterized 28 HER2E cases with whole genomic sequencing (new data). Clinical outcomes (iDFS and RFI for SCAN-B and METABRIC respectively and OS for both) were determined with multi-variate analyses additionally performed. Overall the manuscript is clear and well written. The Discussion attempts to put clinical relevance to the findings and nicely hi-lights limitations of the study. There are some issues of true clinical applicability of identifying this subtype in early stage ER+ breast cancer as currently the use of Prosigna (which does provide subtype distinction) is not commonly used in clinical practice. The results are noteworthy from a genomic perspective of diving deep into this subtype - though is questionable regarding clinical utility at present (in both early and advanced stage disease).

Major Concerns:

- The clinical utility of identifying this ERpHER2n-HER2E cohort (2-6%) is questionable. At present they would be identified as ER positive and HER-2 negative breast cancers. As the authors have shown this subtype is similar to the luminal B subtype with higher proliferation gene expression associated. Furthermore it appears (albeit limited by small sample size) that adjuvant chemotherapy and endocrine therapy may mitigate some of associated poor prognosis. This reviewer would be interested in the authors simulating the recurrence score (Oncotype DX) as they do have RNA expression data to see if the range of recurrence scores is similar or different between ERpHER2n-HER2E and luminal B. As the Oncotype is routine utilized to identify ER+/HER2 negative breast cancer potential benefit from adjuvant chemotherapy – it may be that the use of this currently utilized genomic assay provides sufficient information for clinical actionability (rather than the need to also perform subtype determination).
- The authors have demonstrated that ESR1 expression is lower in ERpHER2n-HER2E relative to luminal A and luminal B. In SCAN-B the study population had ER of at least 10% of greater. In METABRIC the threshold was lower (>1%). As data now supports ER low (less than 10%) should be treated like triple negative breast cancer (with chemotherapy and IO) – it would be more consistent to exclude the ER low cases that were HER2E from the METABRIC study in this combined analyses with SCAN-B.
- The finding of high FGFR4 expression in HER2E tumours is potentially interesting. The referenced study regarding activity of a FGFR4 inhibitor in a HER2E PDX model [Garcia-Recio 2020] adds some evidence for clinical actionability. It may be helpful to define the mechanism of altered expression in HER2E tumours. Is there an activating mutation or CNA in FGFR4? Can it be identified selectively by other methodologies other than PAM50 or RNA expression?

Reviewer #5

(Remarks to the Author)

Version 1:

Reviewer comments:

Reviewer #1

(Remarks to the Author)

With the additional data related to the more mechanistic findings, the paper is significantly improved and has addressed my main concerns. However, it would be helpful to add a more detailed description of the two datasets in the paper (as the authors did in the response to reviewers) to highlight the dataset strengths and weaknesses, particularly with respect to survival using the SCAN-B dataset.

Reviewer #5

(Remarks to the Author)

Reviewer #6

(Remarks to the Author)

The authors have clearly invested significant effort in exploring the molecular landscape of HER2-E within ER+/HER2- breast cancer, but the study may benefit from further refinement to highlight its novelty. Much of the analysis seems to revisit previously established data, with limited advancement toward clinically actionable findings.

Notably, the survival analyses related to their identified biomarkers did not reach statistical significance or show consistent trends across independent datasets. To strengthen the clinical relevance of their findings, the authors linked their molecular signatures to Oncotype DX classifications using RNA-seq. However, this approach may warrant more careful consideration. Oncotype DX is a qPCR-based, clinically validated assay, whereas RNA-seq has not been validated to accurately classify patients into high- or low-risk groups. The methodology used lacks the necessary technical and clinical validation. It would significantly enhance the study if the authors could provide metrics demonstrating the accuracy of their method in predicting

real Oncotype results.

Additionally, the authors may want to temper the significance of their findings and better acknowledge prior research in this field. For example, their results on FGFR have been previously reported, and their exploration of the HER2-E subtype within ER+/HER2- breast cancer largely parallels the influential work of Perou, Prat, Ellis, as well as clinical findings from Dowsett, Turner, and Hayes. It would be valuable for the authors to place their findings in the context of this earlier work and clarify how their results add to the previously published data.

Version 2:

Reviewer comments:

Reviewer #1

(Remarks to the Author)

We appreciate the attempts of the authors to address concerns raised by the reviewers. In particular, it is clear that technical limitations render some of the requests (particularly related to clinical relevance) difficult to provide as requested. Instead the authors chose a different approach, and we agree that this brings the manuscript closer to addressing the concerns raised. Therefore, as the authors have included their new analyses and also more clearly highlighted the remaining concerns as well as acknowledged previous findings, we would agree that the manuscript is now acceptable for publication.

Rebuttal letter NCOMMS-24-15938, Hohmann et al. “Genomic characterization of the HER2-enriched intrinsic molecular subtype in primary ER-positive HER2-negative breast cancer”

We would like to thank the reviewers for their constructive suggestions on how to improve and further clarify our study. We are very grateful for the feedback and comments, which have helped us further strengthen our manuscript. Based on them, we have revised the manuscript as outlined in this rebuttal document, including a substantial number of new analyses and results based both on existing (public) and novel experimental data.

Please find below the detailed point-by-point responses to the reviewers' comments. Reviewers' comments are presented in **bold** and our responses are shown in regular text. Excerpts from the revised version of the manuscript are shown in *italics*. Please note that the actual reference number for a specific literature reference likely differs between the reference list in this document and the revised manuscript version that has been resubmitted. Any page numbers also refer to the revised manuscript in this new submission.

Reviewer #1 (Remarks to the Author): expertise in breast cancer multi-omics

This manuscript focuses on clinicopathological and molecular characterization of a small subset of HER2 enriched tumors that were classified as ER-positive /HER2-negative (ERpHER2n) based upon ER status but, when classified by the PAM50 molecular signature, show up as a small subset of ERp tumors (termed HER2E). These are not considered ERpHER2p or just ERp but rather have an in-between status. The key goal of this paper is to distinguish these tumors from those that are luminal A or B ERp or technically classified as HER2p rather than HER2 enriched using two main datasets, and to use this information to guide future treatment. While the authors do find some statistically significant differences in aspects of the molecular features of this tumor subset, the concern that arises is the clinical significance of these differences particularly as a guide to distinguish this group from other tumors such as luminal B for treatment. The main limitation of this analysis is the bioinformatic reliance on archived data without supplementation using experiment-based validation. Overall, the general conclusions are relatively minimal, citing the immune response and FGFR4 as potentially targetable features without other supporting evidence.

Response:

As a first response to the stated main limitations of the study by the reviewer we have included:

1. More existing, public (“archival”), data to support our results about the immune response / TME in HER2E tumors. Specifically, this includes analysis of matched single cell phenotyped data of >600 METABRIC tumors from Danenberg et al. *Nature Genetics* 2022.
2. New, unpublished novel experimental data including matched DNA methylation analysis (Illumina EPIC beadchips, targeting 800K CpGs) of 499 of our SCAN-B tumors. We use this data to demonstrate a key transcriptional regulatory mechanism of *FGFR4* being hypomethylation of key shore CpGs in the promoter region of *FGFR4*. In normal breast tissue we find that these shore CpGs are hypermethylated. In tumors, this somatic epigenetic switch shows powerful association with high *FGFR4* expression in our breast cancers, and it even also explains heterogeneity of

FGFR4 expression within the PAM50 HER2E subtype (for which *FGFR4* is selected as a prototypical subtype gene by Parker et al. ¹).

3. By extending the analysis above to an additional 645 breast cancers from the TCGA consortium with matched Illumina 450K DNA methylation and RNA-sequencing data comprising all clinical subgroups we demonstrate that this epigenetic *FGFR4* switch is not unique to luminal breast cancer, as it acts as a general mechanism in breast cancer, including also HER2-positive disease, again explaining the heterogeneity of *FGFR4* expression in the malignancy.
4. Finally, we also show that this epigenetic switch for *FGFR4* is also active in human breast cancer cell lines by analysis of 48 cell lines with matched mRNA and DNA methylation data from Iorio et al. (Cell 2016), thereby demonstrating its usability/importance also for forthcoming in vitro studies.
5. To note, the novel global DNA methylation data now associated with this manuscripts likely represents the largest matched DNA methylation / RNA-sequencing ERpHER2n cohort available in the literature at the moment. While we could collect 645 TCGA tumors with RNA-sequencing and DNA methylation data, these include all clinical subgroups. Thus, the actual number of luminal ERpHER2n tumors is less than our novel SCAN-B tumor data (that also has complete clinical data associated with them).

More detailed responses (including new figures) are presented below in response to specific remarks.

Specific comments:

1) As noted by the authors, the aggressive nature of ERpHER2p tumors has been reported previously for patients treated with endocrine therapy. However, their attempts to extend this to patients treated with chemotherapy for overall survival, as they note, was not significant for the SCAN-B dataset but rather showed “a trend towards poor IDFS ... this observation requires validation in larger cohorts due to small patient numbers.” Similarly, the RFI analysis of patients from the METABRIC dataset treated with chemotherapy (CT) and endocrine therapy (ET) is technically significant by p value but limited to 4 HER2E patients out of a total of 72 total. Furthermore, HER2E patients treated with CT and ET did not have significantly different overall survival intervals relative to other luminal subtypes, so the clinical significance is not clear.

Response:

Firstly, we believe it is good to see that we are in agreement with the reviewer about the clinical value of the results for the endocrine treated ERpHER2n patients, and therefore we will not comment further about these results.

The reviewer raises the important concern about clinical significance based on low numbers, stemming from the fact that patients with HER2E tumors only constitute 2-6% of ER+/HER2- breast cancer cases. Despite low case numbers, this patient subgroup represents an unmet clinical need in the management of early ERpHER2n breast cancer, due to the high-risk nature of these HER2E tumors as corroborated by OncotypeDX risk assessment based on RNA-sequencing FPKM data. While we agree with the reviewer about the importance of large sample cohorts and further validations we would first like to emphasize that this study leverages the two largest single-institution cohorts of primary breast cancer with substantial

associated molecular data reported to date (SCAN-B n=4413 ERpHER2n, METABRIC n=1227 ERpHER2n). There is simply, to the best of our knowledge, now other larger appropriate cohorts currently available to validate our findings in, especially for the CT+ET subgroup. TCGA is for instance not an option due to smaller size and would irrespectively not have been an appropriate cohort to use for patient outcome analyses due to biased patient selection and lack of treatment and follow-up data. Similarly, we believe that creating combined, in silico merged, cohorts from many different public studies to reach higher sample numbers would also not be informative due to biases introduced by patient selection / cohort composition, technical platforms, subtyping approaches, differences in ER, PR, HER2 evaluations, etc. As noted in the discussion, we do acknowledge the limitations of the study with respect to patient sizes and we do hope that future validations will support the findings reported here. Forthcoming SCAN-B data releases will be instrumental here, as today >16000 patients have been analyzed.

With this stated, in response to the reviewer's remark that we see as focused on the analysis for the CT+ET patient group specifically we would like to direct the reviewer's attention to:

1. The p-value shown in Figure 2E that the reviewer refers to in the remark (SCAN-B, IDFS, CT+ET) is borderline non-significant (log-rank $p=0.06$), correct = trend. The p-value is a log-rank p-value testing difference between all groups, i.e., the global null hypothesis that all curves are the same. The LumA and LumB curves are obviously not different, while the HER2E patient group stands out as particularly poor. This is actually displayed in the univariate Cox regression data shown in the same plot for both LumB vs LumA (non-significant) and HER2E vs LumA (significant). The reviewer is not commenting about these significant Cox results.
2. Despite that corresponding p-values are significant for CT+ET in METABRIC (log-rank + univariate Cox) we agree with the reviewer that the METABRIC cohort, due to its composition and size is less suited for any conclusions about CT+ET.
3. The corresponding multivariate Cox regression model in the SCAN-B CT+ET cohort using IDFS as endpoint generates a significant p-value and a 95% HR confidence interval not including 1. This supportive, statistically significant, finding is also not mentioned by the reviewer.
4. Neither SCAN-B nor METABRIC are randomized clinical trials. As SCAN-B is a population-representative observational study it reflects national treatment guidelines during the inclusion period, thereby grounding our findings in real-world oncology management practices. Thus, patients do not randomly receive a specific treatment meaning that there are potential competing risks due to age, patient preferences, and comorbidities. Moreover, SCAN-B patients were enrolled up until 2018, meaning that this is still a "young" ERpHER2n cohort with respect to patient outcome as relapses are typically late in this group compared to e.g. TNBC. Finally, even if a patient is diagnosed with a metastasis, modern palliative oncological treatment (relevant for SCAN-B patients) can support patients for a long time. Combined, these aspects have implications for how overall survival can be interpreted.

In the revised manuscript we have made the following changes in the response to this remark (Discussion):

But despite the HER2E subtype making up only a few percent of the ERpHER2n breast cancer patient population, it is important to note that this study leverages the two

largest single-institution cohorts of primary breast cancer with substantial associated molecular data reported to date.

2) Many of the observations (e.g. frequency of HER2-low samples) are inconsistent between the two tumor datasets so it is difficult to determine the clinical significance.

Response:

With respect to this remark we only partially agree with the reviewer. It is correct that the HER2-low frequency differs between SCAN-B and METABRIC as seen in Table 1. As noted, this could be due to technical reasons (e.g. how IHC was performed and evaluated) or cohort composition. In SCAN-B, we based the HER2-low status on reviewed data available in patients charts that were based on routine clinical diagnostic analyses conducted after 2010 (all SCAN-B patients are recruited from September 2010 onwards). METABRIC is a much older cohort for which HER2 assessment was likely not done in routine pathology departments, making it very difficult to assess the robustness and clinical validity of the data. Moreover, as noted the difference (as well as for other clinicopathological variables) could also be due to different cohort composition. As stressed in the discussion of the original manuscript, and as also shown in the publication reporting the used SCAN-B cohort ², the SCAN-B cohort is highly population representative providing support for the generalizability of findings. A similar conclusion / analysis has not been presented for METABRIC. However, for the example highlighted by the reviewer it should be noted that while the frequency of HER2-low differs between cohort, the conclusion becomes the same, i.e., that the HER2E subtype in ERpHER2n tumors is NOT equivalent to the HER2-low subset of patients.

Moreover, in response to the quite general statement of the reviewer (“many of the observations”) without further examples we would like to point out that the agreement between the two cohorts are actually on a molecular level very high in our view. This includes:

1. Clinicopathological characteristics e.g., concerning patient age, tumor grade, and lymph node status
2. Survival outcomes, especially regarding early recurrences for HER2E patients receiving only endocrine therapy
3. Activity of transcriptional programmes as captured by metagenes, including activity of steroid response and immune response programmes
4. Consistency in core gene expression patterns, for example in *ESR1* and *FGFR4*
5. Mutational frequencies of cancer driver genes
6. Genome-wide copy number alterations

We actually believe and argue for that the SCAN-B and METABRIC cohorts support each other very well as a discovery / validation pair, including the newly added single cell data from METABRIC.

3) Molecular features of ERpHER2p like RNA expression and copy number showed biological differences with Luminal A but were not readily distinguishable from Luminal B patients. The networks they identified were not enriched for any pathways and little insight could be gained from transcription factor analysis.

Response:

In response to this remark by the reviewer we would like to stress that the cDEG analysis, copy number analyses, WGS analyses etc. represent small pieces in the puzzle to understand what the HER2E subtype indeed means, and how it differs from LumA and LumB tumors in primary ERpHER2n tumors. Clearly, LumA is different, whereas many of general features of aggressiveness associated with HER2E resembles LumB, but without large distinct differences. We believe that the breast cancer expert reviewer 1 agrees with us that if there would have been major genomic differences (RNA, DNA) these would have been discovered and reported a long time ago considering the >20 years of genomic profiling of breast cancer to date. Their absence can also be attributed to the fact that the PAM50 subtypes themselves do not represent distinct biological classes, but rather a continuum, as previously shown by Veerla et al. ³.

Still, tumors are subtyped differently, and there are clear differences as highlighted including a higher HRD frequency, lower steroid response, a potentially more activated immune response, differences in mutational spectra (e.g. *TP53*) etc. Thus, the genomic analyses are puzzle pieces in the characterisation and understanding of the HER2E subtype that we believe is valuable and important to highlight as some might be treatment predictive in the future.

It should be noted that the cDEGs were obtained from merged results in the SCAN-B and METABRIC cohorts to improve validity. A striking feature here is the complete absence of significantly different expression when comparing ERpHER2n-HER2E to ERpHER2p-HER2E tumors. Together with the HER2-low analysis and the mutational profiling focused on *HER2/ERBB2* we believe these findings reinforce the question of the relevance itself of *HER2/ERBB2* for the HER2E molecular subtype, a key take home from the study in our view (it does not question the treatment predictive / prognostic value of *HER2/ERBB2*).

4) The focus on the immune system was, as noted, limited by inferring immune infiltration using CIBERSORTx rather than direct demonstration.

Response:

We fully agree with the reviewer that the lack of in situ validation of the immune findings was a limitation in the original manuscript, as acknowledged in the original discussion section. Unfortunately, we do not, at the moment, have access to whole slide H&E slides for FFPE tumor sections for the SCAN-B tumors that could be useful for, e.g., TIL evaluation by a pathologist. Neither do we know of any similar data publicly accessible for the METABRIC cohort that could be used for the same purpose.

However, during the review period we have still been able to gather new data to substantiate our findings in response to the reviewer's critical remark. Specifically, we have expanded our study by gathering publicly available single cell data for >600 METABRIC tumors reported by Danenberg et al. ⁴ in 2022. Based on single cell based cell phenotyping data reported by Danenberg et al. we can now show support for higher numbers and proportions of typical immune cells (single cell phenotyped cells) in HER2E tumors compared to LumA and B tumors. The results have been summarized into a new Supplementary Figure S4 associated with the revised manuscript, included below as **Rebuttal Figure 1** for the ease of the reviewer. We believe the addition of this new data addresses the reviewer's remark and also reduces a main limitation of the original study. However, we still acknowledge the need of further in situ validation, e.g. by TIL scoring, and thereby we have opted to not change the

current limitation statement included in the original discussion section.

In response to the reviewer's remark textual changes to the revised manuscript includes:

A new Methods section:

Analysis of Immune Cell Proportions using Single-Cell Phenotype data

The specific cellular composition within HER2E tumors was investigated by using matched in situ single cell estimates of lymphocyte counts derived from single-cell METABRIC data (n=612 patients), as presented by Danenberg et al. ⁵. Using this data, we quantified the counts of different cell phenotypes within each sample, which were then used to calculate the proportions of immune cell types. We performed a comparative analysis of these immune cell counts and proportions across different PAM50 breast cancer subtypes, focusing on identifying patterns in immune cell infiltration.

Results (section HER2E tumors are proliferative and immune-inflamed):

*To complement CIBERSORTx and investigate the specific cellular composition within HER2E tumors, we analyzed in situ estimates of tumor infiltrating lymphocyte counts derived from single-cell METABRIC data, as presented by Danenberg et al. ⁵. After combining immune cell phenotype counts based on Spearman correlation (**Supplementary Figure S4A**) our results indicate that the tumor microenvironment of HER2E tumors tends to be more enriched in immune cells (combined single cell phenotype counts of CD4+ T-cells, CD8+ T-cells, Macrophages, and B-cells) compared to LumA and LumB tumors. The differences in immune cell proportions were more evident when comparing PAM50 subtypes regardless of clinical group (**Supplementary Figure S4B**), as the number of HER2E subtyped ERpHER2n cases was low in the METABRIC subset analyzed by Danenberg et al. (**Supplementary Figure S4C**).*

Discussion:

This was also indicated by the analyses of phenotyped single-cell data obtained from Danenberg et al. ⁵ that highlighted HER2E tumors to exhibit a trend towards a higher enrichment of immune cells (CD4+ T-cells, CD8+ T-cells, Macrophages, and B-cells) compared to LumA and LumB tumors, primarily based on comparisons across clinical groups, necessitated by low sample numbers.

Rebuttal Figure 1. Tumor infiltrating lymphocyte counts derived from single-cell METABRIC data. **A** Spearman correlation heatmap of all single-cell phenotype proportions as originally defined by Danenberg et al. **B** Boxplots showing the phenotype counts of CD4⁺ T-cells, CD8⁺ T-cells, Macrophages, and B-cells across PAM50 subtypes. Additionally, the figure includes boxplots of the combined counts of these immune cell types and the proportion of total cells per sample they represent in each PAM50 subtype. **C** Boxplots showing the phenotype counts of CD4⁺ T-cells, CD8⁺ T-cells, Macrophages, and B-cells across PAM50 subtypes in ERpHER2n breast cancer cases. Additionally, the figure includes boxplots of the combined counts of these immune cell types and the proportion of total cells per sample they represent in each PAM50 subtype. *Statistical significance was assessed using Kruskal-Wallis tests.*

Reviewer #2 (Remarks to the Author): clinical expertise in breast cancer

Comments:

This paper looks at genomic alterations (both DNA and RNA) in a small but distinct subtype (ER positive/HER 2 negative clinically but HER-2 enriched by PAM50) relative to luminal A and luminal B subtypes in two previously published separate cohorts [SCAN-B and METABRIC] of early stage breast cancer treated with older (and not necessarily specified) systemic treatments. The authors have further characterized 28 HER2E cases with whole genomic sequencing (new data). Clinical outcomes (iDFS and RFI for SCAN-B and METABRIC respectively and OS for both) were determined with multi-variate analyses additionally performed. Overall the manuscript is clear and well written. The Discussion attempts to put clinical relevance to the findings and nicely highlights limitations of the study. There are some issues of true clinical applicability of identifying this subtype in early stage ER+ breast cancer as currently the use of Prosigna (which does provide subtype distinction) is not commonly used in clinical practice. The results are noteworthy from a genomic perspective of diving deep into this subtype - though is questionable regarding clinical utility at present (in both early and advanced stage disease).

Response:

We thank the reviewer for these positive comments and have below responded specifically to each remark/concern. This manuscript focuses on primary breast cancer only. To note, we do know the exact specified systemic and endocrine treatments for the clinically reviewed SCAN-B patients as outlined in the Methods section, but not for the METABRIC cohort

Major Concerns:

• The clinical utility of identifying this ERpHER2n-HER2E cohort (2-6%) is questionable. At present they would be identified as ER positive and HER-2 negative breast cancers. As the authors have shown this subtype is similar to the luminal B subtype with higher proliferation gene expression associated. Furthermore it appears (albeit limited by small sample size) that adjuvant chemotherapy and endocrine therapy may mitigate some of associated poor prognosis. This reviewer would be interested in the authors simulating the recurrence score (Oncotype DX) as they do have RNA expression data to see if the range of recurrence scores is similar or different between ERpHER2n-HER2E and luminal B. As the Oncotype is routine utilized to identify ER+/HER2 negative breast cancer potential benefit from adjuvant chemotherapy – it may be that the use of this currently utilized genomic assay provides sufficient information for clinical actionability (rather than the need to also perform subtype determination).

Response:

Similar to a remark by reviewer 1 this remark by reviewer 2 touches upon the important concern about clinical significance based on low numbers, which are stemming from the fact that patients with HER2E tumors only constitute 2-6% of ERpHER2n breast cancer cases. Despite low case numbers, this patient subgroup represents an unmet clinical need in the management of early ERpHER2n breast cancer, due to the high-risk nature of these HER2E tumors. Therefore our in-depth characterization represents an important step to improve therapy and prognoses for these patients.

Here, we would like to emphasize that this study leverages the two largest single-institution cohorts of primary breast cancer with substantial molecular data reported to date (SCAN-B n=4413 ERpHER2n, METABRIC n=1227 ERpHER2n). There is simply, to the best of our knowledge, now other larger appropriate cohorts currently available to target these small subgroups and to validate our findings in, especially for the CT-ET subgroup. TCGA is not e.g. an option and would, irrespectively, not have been an appropriate cohort to use for these analyses due to lack of treatment and follow-up data. Similarly, we believe that creating combined, in silico merged, cohorts from many different studies to reach higher sample numbers would also not be informative due to the large biases introduced by patient selection / cohort composition, technical platforms, subtyping approaches, differences in ER, PR, HER2 evaluations, etc. As stressed in the discussion of the original manuscript, and as also shown in the publication reporting the used SCAN-B cohort ², the SCAN-B cohort is highly population representative providing support for the generalizability of findings, which are thus grounded in real-world oncology management practices. A similar conclusion / analysis has not been presented for METABRIC. As noted in the discussion, we do acknowledge the limitations of the study with respect to patient sizes and we do hope that future validations will support the findings reported here. Forthcoming SCAN-B data releases will be instrumental here, as today >16000 patients have been analyzed.

In the revised manuscript we have made the following changes in the response to the sample size part of this remark (Discussion):

But despite the HER2E subtype making up only a few percent of the ERpHER2n breast cancer patient population, it is important to note that this study leverages the two largest single-institution cohorts of primary breast cancer with substantial associated molecular data reported to date.

Concerning the Oncotype DX we believe this is an excellent suggestion by the reviewer. Based on the *genefu R* package that includes an implementation of the signature we computed scores for our SCAN-B tumors using RNA-sequencing data. Notably, 97% of HER2E tumors had a high score compared to 75% of LumB tumors. If this result is replicated using the clinical assay, all HER2E ERpHER2n tumors would typically be directed to adjuvant chemotherapy and subtyping would not be necessary. This conclusion is likely not that different from if we would have based guidance on Prosigna ROR scores, Ki67 IHC levels (considering the high expression of proliferation related genes in HER2E tumors) combined with e.g. grade and other clinical variables.

However, importantly, neither Oncotype DX nor Prosigna ROR scores inform about alternative treatment strategies. For this we need to understand the molecular basis of the patient/tumor subgroup at hand better – representing a primary objective of the current study.

Textual changes in the revised manuscript with respect to Oncotype DX include:

*Supplementary Methods:

OncotypeDX scores for LumA, LumB and HER2E ERpHER2n tumors were calculated using the Genefu R package (v2.36.0) and classified as either high, intermediate or low-risk cases ⁶. Statistical significance of differences in OncotypeDX scores and classifications were assessed using Mann-Whitney U and Fisher's exact tests, respectively.

*Results (main findings presented in a new Supplementary Table S3, see below):

OncotypeDX classification based on RNA-sequencing FPKM data labelled 97% (86/89) SCAN-B ERpHER2n HER2E tumors as high-risk cases, confirming it to be a high-risk subgroup of breast cancer patients (Supplementary Table S3).

Supplementary Table S3: OncotypeDX scores and risk group classifications of SCAN-B ER+/HER2- cases.

Variable	SCAN-B		
	HER2E (ref)	LumA	LumB
OncotypeDX score	78.8 ± 24	23.3 ± 13.7****	55.6 ± 30.8****
OncotypeDX risk group			
low	0 (0%)	1108 (37%)****	180 (13%)****
intermediate	3 (3%)	1077 (35%)****	167 (12%)****
high	86 (97%)	864 (28%)****	1002 (75%)****

Note: Pairwise comparisons of HER2E vs. LumA and LumB, respectively. Statistics:

Mann-Whitney U test for OncotypeDX score and Fisher's Exact Test for OncotypeDX risk

groups. Significance annotation: * ≤0.05; ** ≤0.01; *** ≤0.001; **** ≤0.0001.

***Discussion:**

In line with our observations, OncotypeDX classification corroborated HER2E tumors to be a high-risk subgroup of ERpHER2n breast cancer patients also according to non-PAM50 based genomic risk assessments.

- **The authors have demonstrated that ESR1 expression is lower in ERpHER2n-HER2E relative to luminal A and luminal B. In SCAN-B the study population had ER of at least 10% of greater. In METABRIC the threshold was lower (>1%). As data now supports ER low (less than 10%) should be treated like triple negative breast cancer (with chemotherapy and IO) – it would be more consistent to exclude the ER low cases that were HER2E from the METABRIC study in this combined analyses with SCAN-B.**

Response:

We fully agree with the reviewer that a change in mindset about ER status is currently ongoing internationally. This is illustrated by e.g. the ASCO/CAP guidelines ⁷ which however still states:

The Expert Panel continues to recommend ER testing of invasive breast cancers by validated immunohistochemistry as the standard for predicting which patients may benefit from endocrine therapy, and no other assays are recommended for this purpose. Breast cancer samples with 1% to 100% of tumor nuclei positive should be interpreted as ER positive. However, the Expert Panel acknowledges that there are limited data on endocrine therapy benefit for cancers with 1% to 10% of cells staining ER positive. Samples with these results should be reported using a new reporting category, ER Low Positive, with a recommended comment. A sample is considered ER negative if < 1% or 0% of tumor cell nuclei are immunoreactive.

As ER IHC scores are not available for the METABRIC cohort we can unfortunately not assess the impact of this remark in the current study. Moreover, the publicly available mRNA data for METABRIC is only available as scaled data, meaning that we in contrast to the RNA-sequenced SCAN-B cohort cannot reliably assess the depth of expression (like FPKM).

Consequently, we cannot reliably in METABRIC make a pseudo cut-off in ESR1 expression to refine the patient set as proposed by the reviewer. In response to the reviewer's remark we have made the following textual changes in the revised manuscript:

Methods:

In SCAN-B, ER-positivity was defined as $\geq 10\%$ of tumor cells being IHC-stained in line with Swedish national guidelines. In the METABRIC cohort ER-positivity was defined as $\geq 1\%$ IHC-stained tumor cells. Due to specific ER IHC scores not being available, reclassifying METABRIC cases according to Swedish national guidelines was not possible.

Discussion:

The observed discrepancy in HER2E prevalence between the SCAN-B and METABRIC cohort is possibly explained by the different definitions of ER-positivity ($\geq 10\%$ in SCAN-B and $\geq 1\%$ in METABRIC) and the SCAN-B cohort being more population representative. Due to specific ER% not being available, we could not reclassify METABRIC cases according to Swedish national guidelines to determine if the ER cut-off level was the main source of the bias between the cohorts.

• The finding of high FGFR4 expression in HER2E tumours is potentially interesting. The referenced study regarding activity of a FGFR4 inhibitor in a HER2E PDX model [Garcia-Recio 2020] adds some evidence for clinical actionability. It may be helpful to define the mechanism of altered expression in HER2E tumours. Is there an activating mutation or CAN in FGFR4? Can it be identified selectively by other methodologies other than PAM50 or RNA expression?

Response:

This is an excellent suggestion by the reviewer on how to improve the current manuscript. There has been additional in vivo reports about FGFR4 inhibitors in the literature (e.g. ⁸ which we do not cite due its focus on anti-HER2 resistance). During the review period of the original manuscript we were able to gather substantial amount of both public but also novel experimental data to address this main remark by the reviewer. Initially, we did not identify a molecular mechanism for the upregulation of *FGFR4* expression by analysis on the DNA level (mutations and CNAs) and therefore decided to expand our study to investigate epigenetic alterations based on DNA methylation profiling. Specifically, we have added substantial new analyses and results to the revised manuscript including novel matched DNA methylation data of 499 SCAN-B tumors (Illumina EPIC v1 beadchips), analysis of 645 TCGA tumors with matched DNA methylation (Illumina 450K beadchips) and RNA-sequencing data of all clinical subgroups and PAM50 subtypes, DNA methylation (Illumina 450K beadchips) analysis of 96 normal breast tissue specimens to demonstrate the putative tissue background methylation state, and analysis of 48 human breast cancer cell lines with matched mRNA (Affymetrix microarrays) and DNA methylation (Illumina 450K beadchips) data.

Specific findings include:

1. *FGFR4* does not appear driven by copy number alterations or somatic alterations as underlined by the total absence of called *FGFR4* amplifications or pathogenic somatic mutations in our WGS-analyzed ERpHER2n HER2E tumors.
2. The new, unpublished novel, experimental data including matched DNA methylation analysis (Illumina EPIC beadchips, targeting 800K CpGs) of 499 of our SCAN-B

tumors demonstrate a key transcriptional regulatory mechanism of *FGFR4* being hypomethylation of key shore CpGs in the promoter region of *FGFR4*. In normal breast tissue we show that these shore CpGs are hypermethylated. In tumors, this somatic epigenetic switch shows powerful association with high *FGFR4* expression in our breast cancers, and it even also explains heterogeneity of *FGFR4* expression within the PAM50 HER2E subtype (for which *FGFR4* is selected as a prototypical subtype gene).

3. By extending the analysis above to an additional 645 breast cancers of all clinical subgroups from the TCGA consortium with matching Illumina 450K DNA methylation data and RNA-sequencing data we demonstrate that this epigenetic *FGFR4* switch is not unique to luminal breast cancer, as it acts as a general mechanism in breast cancer, including also HER2-positive disease, again explaining the heterogeneity of *FGFR4* expression in the malignancy. For the TCGA tumors we took advantage of our developed algorithm to adjust DNA methylation values for tumor cell content, which clearly improved the calling of hypo/hypermethylation status for individual CpGs.
4. Finally, we also show that this epigenetic switch for *FGFR4* is also active in human breast cancer cell lines by analysis of 48 cell lines with matched mRNA and DNA methylation data from Iorio et al. (Cell 2016), thereby demonstrating its usability/importance also for forthcoming in vitro studies.
5. To note, the novel global DNA methylation data now associated with this manuscripts likely represents the largest matched DNA methylation / RNA-sequencing ERpHER2n cohort available in the literature at the moment. While we could collect 645 TCGA tumors with RNA-sequencing and DNA methylation data, these include all clinical subgroups. Thus, the actual number of luminal ERpHER2n tumors is less than our novel SCAN-B tumor data (that also has complete clinical data associated with them).

All results have been summarized into a new main Figure 6 shown below as **Rebuttal Figure 2**, and as a new Supplementary Figure 6 included as **Rebuttal Figure 3** below. We believe these findings substantially increase the novelty of the manuscript. It also opens up an interesting investigation into whether other PAM50 specific genes share a similar on/off regulation.

Large textual changes have been made to describe these results in the revised manuscript. They include:

New Method sections:

DNA methylation analysis of SCAN-B ERpHER2n tumors

DNA methylation analysis was performed on the same DNA used for WGS using Illumina EPIC v1 beadchips according to manufacturer's instructions for 499 tumors. Methylation profiling was performed by the SNP&SEQ Technology Platform in Uppsala (www.genotyping.se). Beta values, representing the level of methylation, were computed in a sample-by-sample context using the minfi R package v1.44 function preprocessNoob() and Infinium probe normalized using the approach described by ⁹.

The Cancer Genome Atlas (TCGA), normal breast tissue, and breast cancer cell line cohorts

Processed and compiled DNA methylation data (Illumina 450K beadchips) and RNA-sequencing FPKM estimates for 645 breast cancers with matched data from the TCGA cohort was obtained from the study by Staaf and Aine ¹⁰. DNA methylation data (beta-values) were

adjusted for tumor cell content as described in ¹⁰ for each tumor. TCGA tumors were PAM50 subtyped using the NCN approach described by Staaf et al. ¹¹, using the same reference sets and code. DNA methylation profiles for 96 normal breast tissue samples analyzed by the Illumina 450K methylation platform were obtained from GSE67919 as processed beta-values ¹². Gene expression data and matched DNA methylation profiles for 48 breast cancer cell lines analyzed by Affymetrix RNA microarrays and Illumina 450K methylation arrays were obtained from the study by Iorio et al. ¹³ as processed RMA expression values and beta-values (GSE68379), respectively.

A new Result section connected to a new main figure and a new supplementary figure (see below):

Epigenetic regulation of the prototypical HER2E subtype gene FGFR4 explains mRNA expression heterogeneity in breast cancer

*FGFR4 was selected to be a prototypical high expressing gene for HER2E in the PAM50 subtyping scheme besides ERBB2 and GRB7 (both located in the minimal amplified region on 17q12) ¹ and has been shown to have an equal or in ERpHER2n disease even slightly larger impact on PAM50 stability than the ERBB2/GRB7 locus ³. Yet, our analyses highlight three important aspects of FGFR4 in breast cancer, i) that heterogeneity in expression within HER2E tumors exists (e.g. Figure 3I), ii) that heterogeneity in expression exists between PAM50 subtypes, with, e.g., high FGFR4 expressing tumors also in the PAM50 LumA and LumB subtypes (**Figure 3I**), and iii) that the gene does not appear driven by copy number alterations or somatic alterations as underlined by the total absence of called FGFR4 amplifications or pathogenic somatic mutations in our WGS-analyzed ERpHER2n HER2E tumors.*

*To better understand FGFR4 expression in our ERpHER2n tumors we generated matched DNA methylation profiles from 499 SCAN-B tumors using Illumina EPIC methylation beadchips (targeting approximately 800K CpGs). Based on clustering of beta-values (representing the level of DNA methylation) for CpGs mapping to the promoter region of FGFR4 we identified a pattern of hypomethylation of CpGs annotated as shores in a subset of cases that showed matched elevated mRNA expression (**Figure 6A** and **Supplementary Figure S6**). In comparison, hypermethylation of these CpGs appear as the background state in normal breast tissue (**Figure 6B**). Importantly, dividing the 499 SCAN-B cases by DNA methylation levels of two key CpGs (cg00618323 and cg20898288) in the shore region stratified tumors distinctively by FGFR4 FKPM expression levels, with tumors showing hypomethylation of these CpGs having markedly elevated mRNA expression (**Figure 6C**). Consistently, this division stratified also ERpHER2n HER2E tumors into a low and high FGFR4 expression group, respectively (**Figure 6C**). Consistent with **Figure 3I** (i.e., presence of high FGFR4 expressing tumors also in other PAM50 subtypes), we found that in all 499 tumors the CpG shore hypomethylated group, while enriched for HER2E tumors, comprised a mix of PAM50 subtypes (**Figure 6D**).*

*To validate these findings and to extend them to breast cancer in general we performed a similar analysis in 645 TCGA breast cancers with matched DNA methylation and RNA-sequencing data. Again, a small subset of tumors with hypomethylation of the shore CpGs (only one shore CpG present in the lower resolution Illumina 450K assay used in TCGA) were identified, which could define DNA methylation subgroups that were strongly correlating with FGFR4 expression in a general breast cancer cohort (**Figure 6E-F**). Similar to SCAN-B, while the hypomethylation cluster was enriched for HER2E tumors, both clusters comprised a mixture of PAM50 subtypes (**Figure 6G**). The DNA methylation-based division of the TCGA*

tumors could also stratify HER2-positive tumors, irrespective of ER-status and PAM50 HER2E status, into high and low FGFR4 expression groups (**Figure 6H**).

Finally, we also analyzed 48 human breast cancer cell lines with matched gene expression and DNA methylation data reported by Iorio et al. ¹³. Similar to the SCAN-B and TCGA tumor cohorts, the DNA methylation status of the cg00618323 shore CpG was again strongly associated with FGFR4 expression levels in the cell line cohort (Wilcoxon's $p=5e-05$, **Supplementary Figure S6**).

New discussion sections / sentences:

While FGFR4 expression is selected to be a defining trait for HER2E tumors it is still heterogeneous, including high expressing tumors of other subtypes but also variability in expression within the HER2E subtype, irrespective of ERBB2 amplification status. This simple illustration warrants further investigation of genes selected to be prototypical drivers of specific PAM50 subtypes. In this study we demonstrate that the key regulator of FGFR4 mRNA expression are not somatic variants or copy number alterations but instead appear to be DNA hypomethylation of shore regions in the promoter of the gene, acting as a switch for FGFR4 expression in breast cancer. Taken together, this mechanism is thus relevant for understanding PAM50 subtyping on a general level, suggesting that a similar mechanism may be relevant also for other PAM50 genes. In more detail, it explains heterogeneity in FGFR4 expression both across PAM50 subtypes and in cell lines, but also within HER2E tumors irrespective of ERBB2 status, and may even be potentially informative as a biomarker for FGFR4 inhibitors.

And

Molecular features that distinctly characterize the HER2E subtype in luminal disease were shown to be consistent with the subtype in HER2-positive disease as well, including a putative epigenetic mechanism for FGFR4 expression in primary breast cancer tissue as well as in cell lines.

Rebuttal Figure 2. Regulation of *FGFR4* expression by promoter shore

hypomethylation in breast cancer. **A** DNA methylation beta values for CpGs associated with *FGFR4* for 499 SCAN-B ERpHER2n tumors analyzed by Illumina EPIC beadchips. Black=Promotor, Gray=shore. Tumors are grouped by beta-values for the two shore CpGs cg00618323 and cg20898288, with a cut-off of 0.6. **B** Beta values for *FGFR4* CpGs in 96 normal breast tissue specimens from GSE67919. **C** Top panel: *FGFR4* FPKM expression in the 499 SCAN-B tumors grouped according to shore CpGs. Bottom panel: *FGFR4* FPKM expression in ERpHER2n HER2E SCAN-B tumors grouped according to shore CpGs. **D** Distribution of PAM50 subtypes in the SCAN-B tumors by CpG status. **E** Top panel, histogram of beta values for cg00618323 in 645 TCGA tumors analyzed by Illumina 450K beadchips. cg20898288 is not present on this array. A cut-off of 0.4 was used define hypomethylation and hypermethylation as the TCGA data was corrected for tumor cell content, thereby removing a considerable impact of normal cell contamination as seen in the beta histogram. Bottom panel: corresponding CpG beta value heatmap for the 645 TCGA tumors. **F** *FGFR4* FPKM expression in the 645 TCGA tumors grouped according to shore CpGs. **G** Distribution of PAM50 subtypes in the TCGA tumors by CpG status. **H** *FGFR4* FPKM expression in HER2-positive tumor subgroups in the TCGA cohort defined by ER-status and HER2E status according to shore CpGs.

Rebuttal Figure 3. DNA methylation analysis of *FGFR4* promoter locus. **A** Heatmap of clustered beta-values for CpGs (rows) in the *FGFR4* promoter locus (defined as upstream and downstream the transcription start site) in 499 SCAN-B tumors (columns) analyzed by the Illumina EPIC v1 assay. Arrow indicates direction of expression. Lower bar plot shows corresponding *FGFR4* FPKM expression values in the tumors. Clusters of tumors with marked higher FPKM expression and apparent hypomethylation of two shore CpGs highlighted in bold are labelled in red. Beta-values are not adjusted for tumor cell content, which means that the beta-value is a mix of tumor cell methylation status and normal cell (background tissue composition) methylation status. **B** Left: histogram of beta-values for the *FGFR4* shore CpG cg00618323 in 48 human breast cancer cell lines. Vertical red line corresponds to a beta-value of 0.4. Center: scatter plot of *FGFR4* gene expression versus the beta-values for cg00618323 in the cell lines. Right: gene expression versus a binary stratification into hypo (beta-value < 0.4) or hypermethylation for cg00618323. *Statistical significance was assessed using Wilcoxon's test.*

REFERENCES FOR THIS REBUTTAL

1. Parker JS, *et al.* Supervised risk predictor of breast cancer based on intrinsic subtypes. *J Clin Oncol* **27**, 1160-1167 (2009).
2. Staaf J, *et al.* RNA sequencing-based single sample predictors of molecular subtype and risk of recurrence for clinical assessment of early-stage breast cancer. *NPJ Breast Cancer* **8**, 94 (2022).
3. Veerla S, Hohmann L, Nacer DF, Vallon-Christersson J, Staaf J. Perturbation and stability of PAM50 subtyping in population-based primary invasive breast cancer. *NPJ Breast Cancer* **9**, 83 (2023).
4. Danenberg E, *et al.* Breast tumor microenvironment structures are associated with genomic features and clinical outcome. *Nat Genet* **54**, 660-669 (2022).
5. Danenberg E, *et al.* Breast tumor microenvironment structures are associated with genomic features and clinical outcome. *Nat Genet* **54**, 660-669 (2022).
6. Gendoo DM, *et al.* Genefu: an R/Bioconductor package for computation of gene expression-based signatures in breast cancer. *Bioinformatics* **32**, 1097-1099 (2016).
7. Allison KH, *et al.* Estrogen and Progesterone Receptor Testing in Breast Cancer: ASCO/CAP Guideline Update. *J Clin Oncol* **38**, 1346-1366 (2020).
8. Zou Y, *et al.* N6-methyladenosine regulated FGFR4 attenuates ferroptotic cell death in recalcitrant HER2-positive breast cancer. *Nat Commun* **13**, 2672 (2022).
9. Holm K, *et al.* An integrated genomics analysis of epigenetic subtypes in human breast tumors links DNA methylation patterns to chromatin states in normal mammary cells. *Breast Cancer Res* **18**, 27 (2016).
10. Staaf J, Aine M. Tumor purity adjusted beta values improve biological interpretability of high-dimensional DNA methylation data. *PLoS One* **17**, e0265557 (2022).
11. Staaf J, *et al.* RNA sequencing-based single sample predictors of molecular subtype and risk of recurrence for clinical assessment of early-stage breast cancer. *npj Breast Cancer* **8**, 94 (2022).
12. Hair BY, *et al.* Body mass index associated with genome-wide methylation in breast tissue. *Breast Cancer Res Treat* **151**, 453-463 (2015).
13. Iorio F, *et al.* A Landscape of Pharmacogenomic Interactions in Cancer. *Cell* **166**, 740-754 (2016).

Rebuttal letter NCOMMS-24-15938A-Z, Hohmann et al. “Genomic characterization of the HER2-enriched intrinsic molecular subtype in primary ER-positive HER2-negative breast cancer”

We would like to thank the reviewers for their constructive suggestions on how to improve and further clarify our study. We are very grateful for the feedback and comments, which have helped us further strengthen our manuscript. Based on them, we have revised the manuscript as outlined in this rebuttal document, including new analyses and results based on SCAN-B RNA-sequencing data.

Please find below the detailed point-by-point responses to the reviewers' comments. Reviewers' comments are presented in **bold** and our responses are shown in regular text. Excerpts from the revised version of the manuscript are shown in *italics*. Please note that the actual reference number for a specific literature reference likely differs between the reference list in this document and the revised manuscript version that has been resubmitted. Any page numbers also refer to the revised manuscript in this new submission.

REVIEWER COMMENTS

Reviewer #1 (Remarks to the Author):

With the additional data related to the more mechanistic findings, the paper is significantly improved and has addressed my main concerns. However, it would be helpful to add a more detailed description of the two datasets in the paper (as the authors did in the response to reviewers) to highlight the dataset strengths and weaknesses, particularly with respect to survival using the SCAN-B dataset.

Response:

We thank the reviewer for this valuable suggestion. With respect to more details, we interpret the reviewer's request as including: a) more details about clinical background data important for tumor classification of the cohorts, b) the cohorts population-representativity, and c) survival aspects. While these were readily available for SCAN-B, publications concerning METABRIC seem to have excluded such information on closer inspection.

With respect to additional clinical background data, we have searched the original METABRIC publication (as well as the later publication by Rueda et al.¹) for more details concerning the ER and HER2 IHC data that could be of value for this study. However, surprisingly, we are not able to locate specific details about these standard biomarkers, i.e. how they were experimentally performed and assessed by the authors. Rueda et al. stated that METABRIC patients had been diagnosed between 1977-2005 in either the UK or Canada, and that they obtained linked pseudo-anonymized clinical data. Curtis et al.² stated that no HER2+ patient received trastuzumab (meaning that there were likely no clinical HER2 IHC/ISH performed in contrast to SCAN-B patients). It is unclear from the publications and associated supplementary data whether the study group re-performed HER2 IHC/ISH, ER assessment, etc, according to relevant guidelines or whether merged clinical data from records were used. Thus, one can only speculate that there could be differences in how these standard biomarkers were assessed. Notably, in the study by Curtis et al. the authors used SNP6 arrays to define HER2-status, not IHC/ISH. Importantly, due to the extensive shortage of technical information about the clinical markers in METABRIC, we have opted not to comment on them in response to this remark to

avoid being speculative. Moreover, it makes reclassification of e.g. ER-status according to Swedish guidelines even more unfeasible. In contrast, for SCAN-B patients, clinicopathological data were extracted from the national breast cancer quality registry which collects data reported by healthcare professionals from the individual pathology and oncology departments into a centralized and standardized database. For >900 patients we performed dedicated clinical review of patient's charts to assure correctness in updated outcome and other information.

In response to the reviewer's comment, we have added the following text (partly reformatted from the previous rebuttal) to address the three main aspects outlined above:

Details about relevant clinical data:

Methods, section *Patient cohorts*

In the SCAN-B cohort HER2-status was determined from routine clinical IHC and FISH analysis as reported in the Swedish national breast cancer quality registry (NKBC). In SCAN-B, ER-positivity was defined as $\geq 10\%$ of tumor cells being IHC-stained in line with Swedish national guidelines. ER and PR data is based on routine clinical IHC and FISH analysis as reported in the NKBC registry. In the METABRIC cohort ER-positivity is expected to be defined as $\geq 1\%$ IHC-stained tumor cells.

Population representativity

Methods, section *Patient cohorts*

In contrast to the used SCAN-B cohort, a comprehensive population representativity analysis for the METABRIC cohort, drawn from a cohort of patients diagnosed between 1977 and 2005 (PMID:30867590), has not been reported.

Survival aspect:

Discussion:

Important to note is that neither SCAN-B nor METABRIC are randomized clinical trials. As SCAN-B is a population-representative observational study it reflects national treatment guidelines during the inclusion period, grounding our findings in real-world oncology management practices. Thus, patients do not randomly receive a specific treatment meaning that there are potential competing risks due to age, patient preferences, and comorbidities that can influence survival analyses. Moreover, modern palliative oncological treatment (relevant for SCAN-B patients) can support patients for considerable time. Combined, these aspects have implications for how overall survival can be interpreted.

Reviewer #5 (Remarks to the Author):

Response:

No comments to respond to.

Reviewer #6 (Remarks to the Author):

The authors have clearly invested significant effort in exploring the molecular landscape of HER2-E within ER+/HER2- breast cancer, but the study may benefit from further refinement to highlight its novelty. Much of the analysis seems to revisit previously established data, with limited advancement toward clinically actionable findings.

Notably, the survival analyses related to their identified biomarkers did not reach statistical significance or show consistent trends across independent datasets.

Response:

The survival analysis performed in this work is focused on the HER2E PAM50 subtype vs PAM50 LumA/LumB, which should be the “biomarkers” the reviewer refers to. These PAM50 subtype classifications were derived through a stringent approach using RNA-sequencing data described in detail in Staaf et al. ³ and validated against commercially available, or research-based equivalent, Prosigna subtype data from matched FFPE tissue in independent cohorts.

We would like to point out that the prognostic findings for the HER2E subtype (poor prognosis) are consistent between SCAN-B and METABRIC (Figure 2, showing Kaplan-Meier plots, univariate, and multivariate Cox regression results). This applies to both ET and CT-ET, with some findings being borderline non-significant due to small group sizes. We believe that it is made clear in the discussion that a major limitation in the interpretation of these outcome findings, despite leveraging the two largest single institution datasets, is the small size of the HER2E group in ERpHER2n tumors, which is due to biology (it is a rare group of tumors in population-representative primary ERpHER2n disease). Notably, the same argument applies also to other studies targeting this rare ERpHER2n subtype.

To reiterate a key response made in the previous rebuttal, to the best of our knowledge there are no other larger, appropriate cohorts currently available for us to further validate our findings, especially for the CT-ET subgroup. The well-known TCGA cohort is, for instance, a poor option for patient outcome analyses due to its smaller size, biased patient selection, lack of treatment data, and insufficient follow-up data. Similarly, we believe that creating a combined cohort by computationally merging different public studies (that have enrolled patients during large and different time spans) to reach higher sample numbers would also not be informative, due to biases introduced by patient selection / cohort composition, technical platforms, subtyping approaches, differences in ER, PR, HER2 evaluations, etc.

To strengthen the clinical relevance of their findings, the authors linked their molecular signatures to Oncotype DX classifications using RNA-seq. However, this approach may warrant more careful consideration. Oncotype DX is a qPCR-based, clinically validated assay, whereas RNA-seq has not been validated to accurately classify patients into high- or low-risk groups. The methodology used lacks the necessary technical and clinical validation. It would significantly enhance the study if the authors could provide metrics demonstrating the accuracy of their method in predicting real Oncotype results.

Response:

The Oncotype DX score analysis (also referred to as the 21-gene recurrence score, RS) was added as a response to a previous reviewer request. In the manuscript we clearly state, as noted by the reviewer, that the scores are computed based on RNA-seq data using the Genefu R package implementation, which uses the terminology “OncotypeDX” ⁴. To avoid confusion,

we have therefore in the revised manuscript changed the naming from “Oncotype DX” to “21-gene recurrence score (RS)” throughout the text.

The reviewer is correct that an accuracy metric of the Genefu “OncotypeDX” algorithm versus the commercial OncotypeDX assay is lacking in the literature. It must, however, be stressed that the request to provide metrics demonstrating the accuracy of the RNAseq-based RS score from Genefu compared to the clinical OncotypeDX assay within this study is associated with several challenging aspects:

1. OncotypeDX was approved for clinical use in Sweden in late 2021, while all SCAN-B patients included in this study were diagnosed and enrolled between Q4 2010 and Q1 2018. Additionally, OncotypeDX is used in some Swedish healthcare regions but not all (including the healthcare region pertinent to this study). Consequently, there is no clinical Oncotype DX data available for any included patient that we can access through patient charts and compare to Genefu’s classification.
2. Regarding us generating the data ourselves, OncotypeDX is performed on FFPE tissue while all RNAseq data available to us was obtained from fresh frozen tissue (i.e., tissue obtained from patients and preserved in RNAlater together with signed informed consent). Thus, any comparison would be performed on different tissue pieces representing an intrinsic source of bias that we could not control for.
3. We do not have access to matched clinical FFPE tissue for SCAN-B patients as this is not automatically collected as research specimens when patients agree to enroll in SCAN-B.
4. Collecting patient FFPE blocks (up to 14 years old for some included patients) that are stored at regional pathology laboratories in our healthcare region to perform the commercial Oncotype DX analysis constitutes a substantial work effort, involving e.g. block identification, physical collection, pathology review, and sectioning. Moreover, as OncotypeDX is a centralized analysis (i.e. tissue samples are sent to a company laboratory located outside a healthcare region), such a project would also require separate new biobank approvals, MTAs, and likely also new ethical permissions, besides considerable funding.
5. Finally, to accurately derive an RNAseq proxy for Oncotype DX, a stringent machine learning scheme is needed. Notably, an RNAseq model that can act both as a single sample predictor and optimally mimic the RS risk classification may be a model that does not necessarily include the recurrence score genes, mimic the original RS equation or the Genefu implementation, or reported cutoffs. In essence, deriving an RNAseq proxy of OncotypeDX represents a comprehensive machine learning study that should investigate multiple different ways of constructing the optimal model.

Taken together, providing a validated RNAseq-based metric comparable to OncotypeDX, though relevant, would represent a major undertaking that is unfortunately not feasible with the material, patient cohort, funding, and timeframe at hand for this submitted project, especially as the OncotypeDX classification is not the main focus of our work.

As an alternative response to the reviewer’s request, we have instead included a new analysis involving a validated RNAseq implementation of the Risk Of Recurrence (ROR) score. ROR

is the risk-score output of the commercially available Prosigna assay, which has been approved for clinical use in multiple countries, including Sweden. Several studies have assessed the agreement in classification between OncotypeDX and Prosigna in head-to-head comparisons, including for instance the study reported by Bartlett et al. ⁵ based on the OPTIMA Prelim Trial. In this study, it was noted that the overall agreement between the two tests (two class stratification into low/intermediate and high-risk) was moderate (kappa 0.44), just as for the comparison between OncotypeDX and the Mammaprint assay. The study by Bartlett et al., however, also hinted towards where most discordance between tests lie, the intermediate risk group, and the transitions between low/intermediate and intermediate/high risks. Notably, when focusing on the group of patients classified as high-risk by OncotypeDX, nearly all of these were also classified as high-risk by Prosigna (who scored almost twice the number of patients as high-risk compared to OncotypeDX) (Figure 1A, Bartlett et al. ⁵).

In contrast to OncotypeDX, in the study by Staaf et al. ³ we developed RNAseq-based single sample prediction models for ROR. These models were subsequently applied to independent RNAseq data from fresh frozen cohorts with available matched FFPE Prosigna ROR scores to validate the approach (thus similar to what the reviewer requests). Based on a pooled cohort analysis in ERpHER2 patients (n=148), it was reported in Staaf et al. that there was an 86% agreement (kappa=0.72) between fresh frozen RNAseq-based ROR risk groups (low/intermediate and high, similar to classes used by Bartlett et al.) and similar ROR risk groups obtained by analysis of matched FFPE tissue (please see ³ for full details).

As the current study uses the deposited RNAseq data from the study by Staaf and coauthors, it means that we can access the RNAseq-based ROR scores for all SCAN-B tumors according to the validated RNAseq model. Thus, in the absence of validated RNAseq-based OncotypeDX scores, we believe the addition of the RNAseq-based ROR scores computed according to the reported validated model, serves as an important alternative response to the reviewer's request for technical validation of RNAseq-based risk scores. Importantly, the conclusion using ROR data is the same as for the RNAseq-based RS score: HER2E tumors are predominantly classified as high-risk (ROR=92%, RS=96% for HER2E tumors with both classifications available), and have significantly higher risk scores and risk score categories than LumA and LumB tumors when tested pairwise. Regarding risk score categories, it should be noted that RNAseq ROR implementation accounts for both tumor size and lymph node status for determination of ROR score and ROR risk cut-offs similar to the original ROR model. Finally, 89.3% of HER2E tumors were classified as high-risk by both signatures.

In response to the reviewer's remark, we have in the revised manuscript included ROR score analysis in a similar way as for the original OncotypeDX analysis. Main changes include:

1. Complete results presented together with the Recurrence Score results in Supplementary Table S3.

Supplementary Table S3: 21-gene Recurrence Score and ROR scores and risk group classifications of SCAN-B ER+/HER2- cases.

Variable	SCAN-B		
	HER2E (ref)	LumA	LumB
21-gene RS	78.4 ± 24.3	23.3 ± 13.6****	55.7 ± 30.8****
21-gene RS risk group			
low	0 (0%)	1079 (36%)****	175 (13%)****
intermediate	3 (4%)	1058 (36%)****	165 (13%)****
High	81 (96%)	842 (28%)****	980 (74%)****
ROR score	73.7 ± 12.6	27.7 ± 14.3****	67.0 ± 10.8****
ROR risk group			
low	0 (0%)	1751 (59%)****	0 (0%)**
intermediate	7 (8%)	816 (27%)****	271 (21%)**
high	77 (92%)	412 (14%)****	1049 (79%)**

Note: Pairwise comparisons of HER2E vs. LumA and LumB, respectively. Statistics:

Mann-Whitney U test for the 21-gene recurrence score (RS)/Risk of Recurrence (ROR)

score and Fisher's Exact Test for 21-gene RS/ROR risk groups. Significance annotation: *

≤0.05; **≤0.01; *** ≤0.001; **** ≤0.0001.

2. Methodological description placed in the Supplementary Methods document.
3. Addition of the following text in the Result section ***HER2E disease is associated with aggressive tumor features relative to LumA:***

Risk of Recurrence (ROR) score classification based on RNA-sequencing derived estimates from Staaf et al.³ also confirmed HER2E tumors to be high-risk cases compared to both LumA and LumB patients in pair-wise comparisons (Supplementary Table S3). Of 84 HER2E tumors having both classifications, 89.3% were classified as high-risk by both gene signatures.

4. Addition of the following text in the Discussion

Consistently, 21-gene Recurrence Score and ROR score classification corroborated HER2E tumors to be a high-risk subgroup of ERpHER2n breast cancer patients. Agreement in risk classification (low/intermediate vs high risk) between commercial breast cancer risk signatures like OncotypeDX, Prosigna, and Mammaprint have been reported to be modest⁵. While the RNA-sequencing based 21-gene Recurrence Score has not been validated versus the commercial implementation (OncotypeDX), the RNA-sequencing ROR scores for SCAN-B tumors have. In a pooled analysis of 148 ERpHER2n patients, an 86% agreement (kappa=0.72) between fresh frozen RNA-sequencing based ROR risk groups (low/intermediate and high) and equivalent ROR risk groups based on matched FFPE tissue was reported³, suggesting that ERpHER2n HER2E tumors would typically be assigned a high-risk classification by commercial risk signatures.

We believe the addition of this new analysis, driven by the important remark by the reviewer, further supports that HER2E tumors at a group level will typically have a high-risk score by (commercial) breast cancer risk predictors, even when estimates are based on RNAseq data.

Additionally, the authors may want to temper the significance of their findings and better acknowledge prior research in this field. For example, their results on FGFR have been previously reported, and their exploration of the HER2-E subtype within ER+/HER2-breast cancer largely parallels the influential work of Perou, Prat, Ellis, as well as clinical findings from Dowsett, Turner, and Hayes. It would be valuable for the authors to place their findings in the context of this earlier work and clarify how their results add to the previously published data.

Response:

We fully agree with the reviewer that seminal findings and studies should be appropriately referenced and highlighted. We would like to point out that we do already reference 15 studies (out of currently >70 references) from the key researchers in the field mentioned by the reviewer, including but not limited to:

- Caldas^{2,6}
- Dowsett^{7,8}
- Perou^{9,10,11,12,13,14}
- Ellis^{8,10,13,15}
- Prat^{9,12,13,14,16,17,18,19}

These citations relate to all aspects of the study, including the background of molecular subtypes in breast cancer, PAM50 subtypes in general, findings about the HER2E subtype specifically, relation of the HER2E subtype with treatment outcomes, data sets, relevant clinical trials, as well as recent FGFR4 therapy experiments in patient-derived organoids.

A primary objective of the current study was to better understand the molecular basis of the HER2E subgroup using multiple omics layers. Thus, what the current study adds to the field is a comprehensive evaluation on multiple omics levels of the HER2E subtype in ERpHER2n disease using the largest single institution multi-omic cohorts reported to date. Some of the specific novelties in comparison to previous studies are (as summarized in the Results and Discussion of the current manuscript):

- Usage of the largest single institution multi-omics cohorts, of which SCAN-B is population representative (diagnosed 2010-2018) with long clinical follow-up for molecular profiling and outcome analysis in primary breast cancer.
- The mainly negative associations of the HER2E subtype with *ERBB2/HER2* as a gene, meaning HER2E is not defined by *ERBB2* mutations, HER2-low status, *ERBB2* overexpression, or clinical misclassification of *ERBB2* gene amplification.
- While a critical predictive/prognostic factor in breast cancer, when considering the above one could provocatively then argue about the role/importance of *ERBB2* in PAM50 subtyping for HER2E tumors. This is substantiated by the similarity of luminal HER2E tumors and ERpHER2p HER2 tumors shown in the study.
- Heterogeneity in *FGFR4* expression, a prototypically selected HER2E subtype gene, that we propose is due to epigenetic regulation applicable to breast cancer in general.

With respect to *FGFR4* specifically, we would like to stress that we do not in any way claim to have “rediscovered” high *FGFR4* expression in HER2E tumors (we do not investigate any other FGFRs) in the current manuscript. While overexpression of *FGFR4* in HER2E tumors is well established (as cited, and due to the design decision in PAM50), the current manuscript

still adds an important new dimension to our understanding of this gene by highlighting a potential epigenetic mechanism of regulation in some, but importantly not all, HER2E tumors.

Understanding FGFR4 regulation is of interest considering recent reports of FGFR4 inhibitors in breast cancer models that we cite (⁹, a study from the lab of C. Perou). In the literature there is surprisingly little data about *FGFR4* and methylation, and very little related to breast cancer. In a recent study published in Nature Communications by Zou et al.²⁰ (not cited in the original manuscript) it was reported that FGFR4 inhibition enhanced susceptibility to anti-HER2 therapy in resistant breast cancer. The mechanistic approach suggested by the authors was that m6A-hypomethylation regulated FGFR4 phosphorylates GSK-3 β and activates β -catenin/TCF4 signaling to drive anti-HER2 resistance ²⁰. Interestingly, it was concluded that based on that *FGFR4* mRNA expression remained unchanged after treatment with 5-azacytidine (a DNA methylation inhibitor) or trichostatin A (HDAC inhibitor) in two cell lines, (SK-BR-3 and MDA-MB-361) neither DNA methylation nor histone acetylation was the reason for *FGFR4* upregulation. Our results in hundreds of primary tumor tissue samples clearly suggest otherwise. We cannot help to notice that untreated MDA-MB-361 cells (MDA-MB-361, but not SK-BR-3, is included among the 48 cell lines from Iorio et al.²¹ that we investigate) show hypomethylation of the specific shore annotated CpG and high *FGFR4* expression (Rebuttal Figure 1, below). If the regulatory mechanism is as we propose, 5-aza treatment (demethylation) would have no effect on *FGFR4* expression in this cell line (which is also what the authors report), as the CpG site(s) is already hypomethylated (and thus cannot become more hypomethylated by 5-aza). Together, this may suggest that the authors may have reached a potentially wrong conclusion about the role of DNA methylation by overlooking the background state of a regulatory site in the promoter region. Considering this observation, we have chosen to not cite the study by Zou et al. due to the uncertainties in the baseline epigenetic state of the cells they report.

Rebuttal Figure 1. *FGFR4* mRNA expression and beta value of key shore CpG present in Illumina 450K DNA methylation arrays for 48 breast cancer cell lines.

In response to the reviewer's remark, we have made textual changes to the Discussion section of the manuscript to emphasize even more the consistency between specific results of our study and previous key studies in the field, such as:

Corroborating previous studies, HER2E tumors were associated with clinical features of aggressive disease akin to LumB such as high NHG and nodal positivity when compared to LumA tumors. As also reported previously, the HER2E subtype was associated with poor IDFS and OS in patients treated with endocrine therapy^{7, 15, 17, 19, 22}.

And:

Consistently, 21-gene Recurrence Score and ROR score classification corroborated HER2E tumors to be a high-risk subgroup of ERpHER2n breast cancer patients.

Furthermore, we have shortened the Discussion section by removing text related to findings that could be viewed as confirmative of previous studies, e.g. about aggressive features of LumB and HER2E tumors. We have also made attempts to further accentuate corroborative findings and findings from our primary tumors that are consistent with reports in metastatic disease (typically in smaller clinical trial cohorts) by small textual changes in several sentences. We believe the modified Discussion section better complies with the reviewer's request for a more tempered discussion of our findings in light of previous publications within the breast cancer field.

References for this rebuttal

1. Rueda OM, *et al.* Dynamics of breast-cancer relapse reveal late-recurring ER-positive genomic subgroups. *Nature* **567**, 399-404 (2019).
2. Curtis C, *et al.* The genomic and transcriptomic architecture of 2,000 breast tumours reveals novel subgroups. *Nature* **486**, 346-352 (2012).
3. Staaf J, *et al.* RNA sequencing-based single sample predictors of molecular subtype and risk of recurrence for clinical assessment of early-stage breast cancer. *NPJ Breast Cancer* **8**, 94 (2022).
4. Gendoo DM, *et al.* Genefu: an R/Bioconductor package for computation of gene expression-based signatures in breast cancer. *Bioinformatics* **32**, 1097-1099 (2016).
5. Bartlett JM, *et al.* Comparing Breast Cancer Multiparameter Tests in the OPTIMA Prelim Trial: No Test Is More Equal Than the Others. *J Natl Cancer Inst* **108**, (2016).
6. Danenberg E, *et al.* Breast tumor microenvironment structures are associated with genomic features and clinical outcome. *Nat Genet* **54**, 660-669 (2022).
7. Bergamino MA, *et al.* HER2-enriched subtype and novel molecular subgroups drive aromatase inhibitor resistance and an increased risk of relapse in early ER+/HER2+ breast cancer. *EBioMedicine* **83**, 104205 (2022).
8. Griffith OL, *et al.* The prognostic effects of somatic mutations in ER-positive breast cancer. *Nat Commun* **9**, 3476 (2018).

9. Garcia-Recio S, et al. FGFR4 regulates tumor subtype differentiation in luminal breast cancer and metastatic disease. *J Clin Invest* **130**, 4871-4887 (2020).
10. Parker JS, et al. Supervised risk predictor of breast cancer based on intrinsic subtypes. *J Clin Oncol* **27**, 1160-1167 (2009).
11. Perou CM, et al. Molecular portraits of human breast tumours. *Nature* **406**, 747-752 (2000).
12. Prat A, et al. Molecular features and survival outcomes of the intrinsic subtypes within HER2-positive breast cancer. *J Natl Cancer Inst* **106**, (2014).
13. Prat A, et al. Concordance among gene expression-based predictors for ER-positive breast cancer treated with adjuvant tamoxifen. *Ann Oncol* **23**, 2866-2873 (2012).
14. Cejalvo JM, et al. Clinical implications of the non-luminal intrinsic subtypes in hormone receptor-positive breast cancer. *Cancer Treat Rev* **67**, 63-70 (2018).
15. Ellis MJ, et al. Randomized phase II neoadjuvant comparison between letrozole, anastrozole, and exemestane for postmenopausal women with estrogen receptor-rich stage 2 to 3 breast cancer: clinical and biomarker outcomes and predictive value of the baseline PAM50-based intrinsic subtype--ACOSOG Z1031. *J Clin Oncol* **29**, 2342-2349 (2011).
16. Prat A, et al. Everolimus plus Exemestane for Hormone Receptor-Positive Advanced Breast Cancer: A PAM50 Intrinsic Subtype Analysis of BOLERO-2. *Oncologist* **24**, 893-900 (2019).
17. Prat A, et al. Prognostic Value of Intrinsic Subtypes in Hormone Receptor-Positive Metastatic Breast Cancer Treated With Letrozole With or Without Lapatinib. *JAMA Oncol* **2**, 1287-1294 (2016).
18. Schettini F, et al. Clinical, pathological, and PAM50 gene expression features of HER2-low breast cancer. *NPJ Breast Cancer* **7**, 1 (2021).
19. Prat A, et al. Correlative Biomarker Analysis of Intrinsic Subtypes and Efficacy Across the MONALEESA Phase III Studies. *J Clin Oncol* **39**, 1458-1467 (2021).
20. Zou Y, et al. N6-methyladenosine regulated FGFR4 attenuates ferroptotic cell death in recalcitrant HER2-positive breast cancer. *Nat Commun* **13**, 2672 (2022).
21. Iorio F, et al. A Landscape of Pharmacogenomic Interactions in Cancer. *Cell* **166**, 740-754 (2016).

22. Dunbier AK, *et al.* Association between breast cancer subtypes and response to neoadjuvant anastrozole. *Steroids* **76**, 736-740 (2011).